# On the Mechanisms of Weak-to-Strong Generalization: A Theoretical Perspective

**Behrad Moniri**
University of Pennsylvania
Philadelphia, PA
bemoniri@seas.upenn.edu

**Hamed Hassani**
University of Pennsylvania
Philadelphia, PA
hassani@seas.upenn.edu

## Abstract

Weak-to-strong generalization—where a student model trained on imperfect labels generated by a weaker teacher nonetheless surpasses that teacher—has been widely observed, but the mechanisms that enable it have remained poorly understood. In this paper, through a theoretical analysis of simple models, we uncover three core mechanisms that can drive this phenomenon. First, by analyzing ridge linear regression, we study the interplay between the teacher and student regularization parameters and prove that a student can compensate for a teacher's under-regularization and achieve lower test error. We also analyze the role of the parameterization regime of the models and show that qualitatively different phenomena can happen in different regimes. Second, by analyzing weighted ridge linear regression, we show that a student model with a regularization structure better aligned to the target function, can outperform its teacher. Third, in a non-linear multi-index learning setting, we demonstrate that a student can learn easy, task-specific features from the teacher while leveraging its own broader pre-training to learn hard-to-learn features that the teacher cannot capture.

## 1 Introduction

Weak-to-strong generalization refers to the phenomenon where a strong (student) model trained on data produced by a weak (teacher) model can sometimes significantly surpass the teacher's performance. This concept was first introduced by Burns et al. [2024], where the authors fine-tuned the GPT-2 model [Radford et al., 2019] (the teacher) for a specific task using ground-truth labels, subsequently employing the fine-tuned model to generate synthetic samples for the same task. These synthetic samples were then used to fine-tune GPT-4 [Achiam et al., 2023] (the student). Remarkably, the fine-tuned student model outperformed its teacher in certain settings despite having access only to the imperfect synthetic data generated by the teacher.

Weak-to-strong generalization is an especially important phenomenon from a practical perspective because of its implications for the emerging question of *superalignment* [OpenAI, 2023]; i.e., can humans steer models with potentially superhuman capabilities to become aligned to human norms and values [Burns et al., 2024]? Considering the weak model as a proxy for humans, the possibility of the weak-to-strong generalization phenomenon suggests that the answer can be affirmative.

Despite its practical importance, the mechanisms that enable weak-to-strong generalization are still not fully understood. Regularization has empirically been shown to play a critical role in enabling weak-to-strong generalization. However, despite recent theoretical progress demonstrating that regularizing the student is necessary in some settings [Medvedev et al., 2025], the full picture of the effects and the interplay of the regularization of both the student and teacher models in weak-to-strong generalization is still unclear. For example, prior work on weak-to-strong generalization mainly focus on ridgeless regression (see e.g., Dong et al. [2025], Xue et al. [2025], Ildiz et al. [2025], etc.).

39th Conference on Neural Information Processing Systems (NeurIPS 2025).

Additionally, prior work assumes that the teacher and student models have frozen representations, and only a linear head is trained through a convex objective (see e.g., Ildiz et al. [2025], Medvedev et al. [2025], Dong et al. [2025], Xue et al. [2025], Charikar et al. [2024], etc.). However, fine-tuning can in practice go beyond this linearized regime and update model features as well. Burns et al. [2024] empirically demonstrated that updating the features yields substantially stronger weak-to-strong gains than only updating a linear head. For these cases, a linearized theoretical model might not suffice to capture all the relevant phenomena. This motivates a theoretical study beyond the linearized regimes and an analysis of the role of feature learning on weak-to-strong generalization.

To take steps towards better understanding these aspects of training on weak-to-strong generalization, in this paper, we conduct a thorough theoretical study of this phenomenon in prototypical theoretical models. For the linear setting, we let the student and the teacher be high-dimensional standard and weighted ridge regression models and study how the explicit regularizations of the teacher and student models affect weak-to-strong generalization. We also investigate the role of the parameterization regime of the models. As the choice of regularization, we consider ridge and weighted ridge penalties. For the nonlinear case, we consider the problem of learning from multi-index models where the models learn relevant features through a non-convex optimization objective. By studying this setting, we characterize how knowledge propagates between the models.

## 1.1 Contributions

Here we discuss the main contributions of the paper. We characterize three mechanisms that can enable weak-to-strong generalization.

- In Section 2.1, we consider a setting where the student and the teacher are trained with ridge regression. We fully characterize the test error of the models in the high-dimensional proportional regime by deriving asymptotic expressions for the test errors. Using these expressions, we study the conditions where the student model outperforms the teacher. We show that the student model can outperform the teacher by *adequately compensating the under-regularization of the teacher*. We further prove that different parameterization regimes of the student model can result in qualitatively different phenomena.

- In Section 2.2, we consider a setting where the teacher is again trained with ridge regression. However, we train the student with a *weighted* ridge regularization. We again fully characterize the limiting test errors of the models in the high-dimensional proportional limit, and show that weak-to-strong generalization can happen *when the regularization structure of the student is better suited for the task*.

- In Section 3, we study learning form a nonlinear multi-index learning function that can be decomposed to a mix of *easy-* and *hard*-to-learn components by applying a single step of gradient descent on the first layer weights of two-layer neural networks. We assume that the easy component is highly specialized and task-specific, but, the hard component is a component shared across many tasks. We consider a case where the strong student model has already learned the hard component during pre-training, but not the easy (yet task-specific) component. We show that even if the teacher model is not able to learn the hard components on its own, a pre-trained student can learn the easy component from the teacher while still *retaining* the knowledge from pre-training for the hard component.

## 1.2 Related Works

The machine learning community has shown growing interest in weak-to-strong generalization. In this section, we review these results.

**Theoretical Results.**   Prior work examines scenarios in which both the student and teacher rely on fixed, pre-trained feature representations. Wu and Sahai [2024] analyze a stylized classification task under an over-parameterized spiked-covariance model with Gaussian covariates, where the teacher model does not have the capability to fit the target function, and the student has a structure that is better aligned with the target. Ildiz et al. [2025] investigate weak-to-strong generalization for high-dimensional ridgeless regression. Building on this line and in a similar setting, Dong et al. [2025], Xue et al. [2025] study how mismatches between student and teacher features affect generalization: Dong et al. [2025] focus on a ridgeless, variance-dominated linear regime in which both models have negligible bias and show that weak-to-strong transfer occurs when the student's features have lower intrinsic dimension. Xue et al. [2025] consider the same setting and propose that the overlap

between the subspace of features that teacher model has not learned, and the subspace of features that the student model has learned during pre-training govern weak-to-strong generalization. In contrast, this paper analyzes linear models in the high-dimensional proportional regime, covering both under- and over-parameterized cases, where models can have large bias. We also explicitly investigate the role of regularization on weak-to-strong generalization. Complementing these model-centric studies, Shin et al. [2025] take a data-centric approach and study the aspects of data enable weak-to-strong generalization.

Relatedly, Medvedev et al. [2025] consider two-layer neural networks with random first-layer weights (random-features models) and show that, when the student is much wider than the teacher, early stopping is essential for weak-to-strong generalization. However, they assume the teacher is already optimally trained and do not analyze the role of its training. Charikar et al. [2024] propose that the erroneous knowledge that the strong model does not obtain from the weak model characterizes how much the strong model improves over the weak model.

**Empirical Studies.** Following the pioneering work of Burns et al. [2024], different variants and applications of weak-to-strong generalization have been studied. Bansal et al. [2025], Yang et al. [2024] let the weak model generate data with chain-of-thought to supervise the student models. Ji et al. [2024], Tao and Li [2024] use weak-to-strong generalization for the problem of alignment. Guo et al. [2024] study this phenomenon in vision foundation models. Liu and Alahi [2024] propose a hierarchical mixture of experts method to boost weak-to-strong generalization. Mulgund and Pabbaraju [2025] characterize the gain in performance of the student model over the teacher model in terms of the misfit between the models.

## 1.3 Notation

We denote vector quantities by **bold** lower-case, and matrix quantities by **bold** upper-case. We use $\|\cdot\|_{\mathrm{op}}$, $\|\cdot\|_{\mathrm{Fr}}$ to denote the operator (spectral) and Frobenius norms. Given an indexed set of vectors $\{\mathbf{x}_i\}_{i=1}^n$, we use the upper case to denote the (row-wise) stacked matrix, e.g. $\mathbf{X} \triangleq [\mathbf{x}_1 \quad \cdots \quad \mathbf{x}_n]^\top$. Throughout the paper, we use the standard asymptotic notation $o(\cdot), O(\cdot), \Omega(\cdot), \Theta(\cdot)$. Finally, we use $\to_{\mathbb{P}}$ to denote convergence in probability.

## 2 The Linearized Case

During the fine-tuning of pre-trained large-scale models, the training dynamic often falls into a kernel regime where the features are not evolved [Wei et al., 2022, Malladi et al., 2023]. Motivated by these observations, in this section we cast the fine-tuning problem as a linear regression problem over Gaussian features. We aim to analyze the role of student and teacher *regularization*, and also the *parameterization regimes* of the models in weak-to-strong generalization.

Assume that the teacher model has access to $n_t$ independent samples $\mathcal{S}_t = \{(\mathbf{x}_i, y_i)\}_{i=1}^{n_t}$ drawn according to

$$\mathbf{x}_i \sim \mathsf{N}(\mathbf{0}, \mathbf{I}_{\mathrm{d_x}}), \quad y_i = \boldsymbol{\beta}_\star^\top \mathbf{x}_i + \varepsilon_i \tag{1}$$

where $\boldsymbol{\beta}_\star \in \mathbb{R}^{\mathrm{d_x}}$ is an unknown target vector, and $\varepsilon_i \sim \mathsf{N}(0, \sigma_\varepsilon^2)$ is an independent additive noise. The teacher model $\hat{f}_t : \mathbb{R}^{\mathrm{d_x}} \to \mathbb{R}$ is fit on the features $\{\mathbf{x}_i\}_{i=1}^{n_t}$ using these labeled samples. The teacher is then used to generate synthetic labels for $n_s \in \mathbb{N}$ unlabeled covariates $\mathcal{S}_s = \{\tilde{\mathbf{x}}_i\}_{i=1}^{n_s}$ drawn independently from the same distribution according to $\tilde{\mathbf{x}}_i \sim \mathsf{N}(\mathbf{0}, \mathbf{I}_{\mathrm{d_x}})$ as $\tilde{y}_i = \hat{f}_t(\tilde{\mathbf{x}}_i)$. These samples are then used to train the student model $\hat{f}_s : \mathbb{R}^{\mathrm{d_x}} \to \mathbb{R}$.

We focus on the following two settings, each showcasing a different mechanism that can enable weak-to-strong generalization.

**Setting 1: Ridge Regression.** We train the teacher $\hat{f}_t(\mathbf{x}) = \hat{\boldsymbol{\beta}}_t^\top \mathbf{x}$ and the student $\hat{f}_s(\mathbf{x}) = \hat{\boldsymbol{\beta}}_s^\top \mathbf{x}$ using (standard) ridge regression. We prove that a properly regularized student can outperform the teacher, in the case where the regularization parameter of the teacher is set to be smaller that the optimal regularization parameter. This is an example of weak-to-strong generalization through *adequately compensating under-regularization*. Furthermore, we show that *two qualitatively different scenarios* can arise depending whether the student model is *over- or under-parametrized*. This aligns with Burns et al. [2024] and Medvedev et al. [2025], which show that student regularization is necessary for weak-to-strong generalization; we extend their work by analyzing the role of teacher

regularization, and a finer-grained analysis of student regularization and model parameterization, revealing new phenomena.

**Setting 2: Weighted Ridge Regression.** We train $\hat{f}_t(\mathbf{x}) = \boldsymbol{\beta}_t^\top \mathbf{x}$ using (standard) ridge regression and $\hat{f}_s(\mathbf{x}) = \boldsymbol{\beta}_s^\top \mathbf{x}$ using weighted ridge regression [Hoerl and Kennard, 1970, Casella, 1980]. We show that the strong student model can *leverage better regularization structure* and outperform the weak teacher, *even if* the regularization parameter for the teacher is tuned optimally. We argue that a student model can have a more suitable regularization either by using an architecture that is better tailored to the task or by benefiting from more effective pre-training.

We consider growing $n_s, n_t, \mathrm{d_x}$ following the high-dimensional limit. Although our results are proven for this asymptotic regime, through numerical experiments, we show that they still match simulations very well, even for moderately large values of $n_s, n_t, \mathrm{d_x}$.

**Assumption 1.** *Assume that $n_t, n_s$ and $\mathrm{d_x}$ all tend to infinity with a proportional rate; i.e.,*

$$\mathrm{d_x}/n_s \to \gamma_s > 0, \quad \text{and} \quad \mathrm{d_x}/n_t \to \gamma_t > 0.$$

In this high-dimensional limit, we characterize the test errors achieved by the teacher and student models given by

$$\mathcal{L}_\mathrm{t} = \mathsf{E}_{\mathbf{x},y}\left(y - \hat{\boldsymbol{\beta}}_t^\top \mathbf{x}\right)^2 = \sigma_\varepsilon^2 + \|\hat{\boldsymbol{\beta}}_t - \boldsymbol{\beta}_\star\|_2^2 \tag{2}$$

$$\mathcal{L}_\mathrm{s} = \mathsf{E}_{\mathbf{x},y}\left(y - \hat{\boldsymbol{\beta}}_s^\top \mathbf{x}\right)^2 = \sigma_\varepsilon^2 + \|\hat{\boldsymbol{\beta}}_s - \boldsymbol{\beta}_\star\|_2^2$$

where $(\mathbf{x}, y)$ is an independent test sample drawn from (1). We then use these characterizations to study the conditions under which the student model outperforms the teacher.

## 2.1 Setting 1: High-Dimensional Ridge Regression

In this section, we assume that the teacher fits a linear regression model $\hat{f}_t(\mathbf{x}) = \hat{\boldsymbol{\beta}}_t^\top \mathbf{x}$ trained on the samples $\mathcal{S}_t$, and is given by

$$\hat{\boldsymbol{\beta}}_t = \operatorname*{argmin}_{\boldsymbol{\beta} \in \mathbb{R}^{\mathrm{d_x}}} \left[ \frac{1}{n_t} \sum_{(\mathbf{x}_i, y_i) \in \mathcal{S}_t} \left( y_i - \boldsymbol{\beta}^\top \mathbf{x}_i \right)^2 + \lambda_t \|\boldsymbol{\beta}\|_2^2 \right] \tag{3}$$

where $\lambda_t \in \mathbb{R}$ is the teacher ridge regularization parameter. The student is also a linear model $\hat{f}_s(\mathbf{x}) = \hat{\boldsymbol{\beta}}_s^\top \mathbf{x}$ trained on fresh samples $\mathcal{S}_s$ labeled by the teacher model, and is given by

$$\hat{\boldsymbol{\beta}}_s = \operatorname*{argmin}_{\boldsymbol{\beta} \in \mathbb{R}^{\mathrm{d_x}}} \left[ \frac{1}{n_s} \sum_{\tilde{\mathbf{x}}_i \in \mathcal{S}_s} \left( \boldsymbol{\beta}^\top \tilde{\mathbf{x}}_i - \hat{\boldsymbol{\beta}}_t^\top \tilde{\mathbf{x}}_i \right)^2 + \lambda_s \|\boldsymbol{\beta}\|_2^2 \right] \tag{4}$$

in which $\lambda_s \in \mathbb{R}$ is the student regularization parameter. We characterize the test error of these models in the high-dimensional proportional limit of Assumption 1. Our characterization of the test errors $\mathcal{L}_s, \mathcal{L}_t$ will be in terms of the following quantities from the random matrix theory literature (see e.g., Bai and Silverstein [2010]).

**Definition 2.** *Let $m(\lambda; \gamma)$ be the Stieltjes transform of the Marchenko-Pastur law with parameter $\gamma$ evaluated at $-\lambda$; i.e.,*

$$m(\lambda; \gamma) = \int \frac{\mathrm{d}\mu_{\mathrm{MP}(\gamma)}(s)}{s + \lambda} = -\frac{1}{2\gamma\lambda}\left[1 - \gamma + \lambda - \sqrt{(1 + \gamma + \lambda)^2 - 4\gamma}\right].$$

*Also, for $p \in \{s, t\}$, we define $m_{p,1} = m(\lambda_p, \gamma_p)$ and $m_{p,2} = -\frac{\partial m}{\partial \lambda}\big|_{\lambda_p, \gamma_p}$.*

The test error of $\hat{\boldsymbol{\beta}}_t$ in the high-dimensional proportional limit has been studied extensively in the literature [Tulino and Verdú, 2004, Dobriban and Wager, 2018, Hastie et al., 2022]. The following proposition characterizes the test error of $\hat{\boldsymbol{\beta}}_t$ in our setting.

**Proposition 3.** *Under the condition that $\boldsymbol{\beta}_\star \sim \mathsf{N}(\mathbf{0}, \mathrm{d_x}^{-1}\mathbf{I}_{\mathrm{d_x}})$ independent of other sources of randomness in the problem, in the high-dimensional proportional limit of Assumption 1, we have*

$$\mathcal{L}_t \to_\mathbb{P} \sigma_\varepsilon^2 + \left(\lambda_t - \sigma_\varepsilon^2 \gamma_t\right)\lambda_t m_{t,2} + \sigma_\varepsilon^2 \gamma_t m_{t,1},$$

*where $m_{t,1}$ and $m_{t,2}$ are defined in Definition 2.*

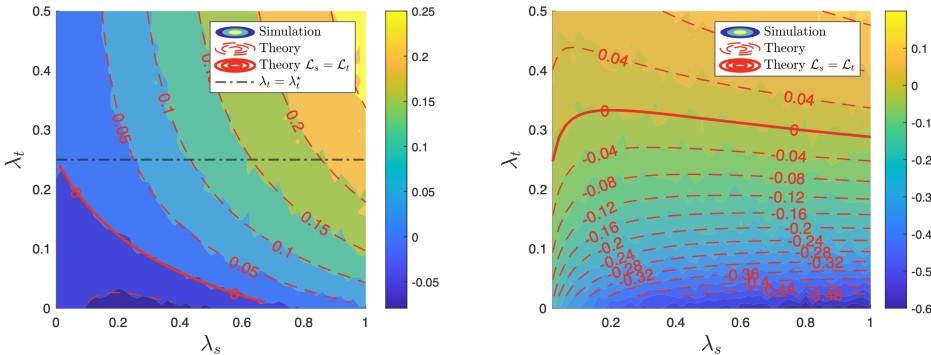

Figure 1: Test-error difference $\mathcal{L}_s - \mathcal{L}_t$ as a function of $(\lambda_t, \lambda_s)$ in the setting of Section 2.1. Filled contours are numerical simulations, and the dashed red contours follow the expressions of Theorem 4. The solid curve marks $\mathcal{L}_s = \mathcal{L}_t$, and the dashed black curve is $\lambda_t = \lambda_t^\star$. **Left**: under-parameterized student. **Right**: over-parameterized student. See Section 4 for more details.

In the following theorem, we study test error of the student model $\hat{\boldsymbol{\beta}}_s$.

**Theorem 4.** *Under the same assumptions as Proposition 3, the test errors of $\hat{\boldsymbol{\beta}}_s$ and $\hat{\boldsymbol{\beta}}_t$ satisfy*

$$\mathcal{L}_s - \mathcal{L}_t \to_{\mathbb{P}} \Delta := (\sigma_\varepsilon^2 \gamma_t - \lambda_t) \left[ (m_{t,1} - \lambda_t m_{t,2}) \left( \lambda_s^2 m_{s,2} - 2\lambda_s m_{s,1} \right) \right] + \lambda_s^2 m_{s,2} \left( 1 - \lambda_t m_{t,1} \right).$$

*where $m_{t,1}, m_{t,2}, m_{s,1}, m_{s,2}$ are defined in Definition 2.*

This theorem fully characterizes the limit of the test error of the student model in the high dimensional proportional limit. The formulas for the limiting errors derived in this theorem can be used to make numerical predictions for $\mathcal{L}_s - \mathcal{L}_t$. Figure 1 shows an example, supporting that the theoretical predictions of Theorem 4 match very well with simulations even for moderately large $d, n_s, n_t$. See Section 4 for more details on the experimental settings. We use this theorem to study the test error of the models as a function of overparamterization in Section E.

In the next theorem, we use the formula for the limiting value of $\mathcal{L}_s - \mathcal{L}_t$ from Theorem 4 to study the conditions on $\gamma_s, \gamma_t, \lambda_s, \lambda_t, \sigma_\varepsilon^2$ under which the student model outperforms the teacher; i.e., the conditions of weak-to-strong generalization.

**Theorem 5.** *Under the conditions of Theorem 4, the (limiting) test errors of the student and teacher models satisfy the following:*

- *If $\lambda_t \geq \sigma_\varepsilon^2 \gamma_t$, we have $\mathcal{L}_s \geq \mathcal{L}_t$.*

- *If $\lambda_t < \sigma_\varepsilon^2 \gamma_t$, two cases can happen:*

  - *If $0 < \gamma_s < 1$, there exists $\bar{\lambda} \geq 0$ such that $\mathcal{L}_s < \mathcal{L}_t$ for all $\lambda_s \in (0, \bar{\lambda})$.*

  - *If $\gamma_s > 1$ and if the parameters $\gamma_t, \gamma_s, \lambda_t, \sigma_\varepsilon$ satisfy*

$$\frac{\lambda_t - \gamma_t \sigma_\varepsilon^2}{\sqrt{(1 + \gamma_t + \lambda_t)^2 - 4\gamma_t}} > \frac{1}{1 - 4\gamma_s - 4\sqrt{\gamma_s^2 - \gamma_s}}, \tag{5}$$

    *then there exists $\bar{\lambda}_-, \bar{\lambda}_+ \geq 0$ such that $\mathcal{L}_s < \mathcal{L}_t$ for all $\lambda_s \in (\bar{\lambda}_-, \bar{\lambda}_+)$. Moreover, if (5) does not hold, we have $\mathcal{L}_s \geq \mathcal{L}_t$.*

Note that under the setting of this section, the optimal ridge regularization parameter for the weak model is known to be equal to $\lambda_t^\star = \sigma_\varepsilon^2 \gamma_t$ [Dobriban and Wager, 2018, Theorem 2.1]. Theorem 5 states that if the teacher is *over-regularized* ($\lambda_t \geq \lambda_t^\star$), the student can never outperform it. In the case that the teacher is *under-regularized* ($\lambda_t < \lambda_t^\star$), the parameterization regime of the student model $\gamma_s$ plays a key role. In particular, if $\gamma_s < 1$ (i.e., the student is *under-parameterized*), the student model can outperform the teacher by further regularization as long as $0 < \lambda_s < \bar{\lambda}$. However, if $\gamma_s > 1$ (i.e., the student in *over-parameterized*), as long as (5) holds, $\lambda_s$ should be larger than a certain threshold for it to outperform the teacher. Otherwise, the student will always have a worse performance compared to the teacher.

The phase transitions predicted in Theorem 5 can be seen in Figure 1, where for each $(\lambda_t, \lambda_s)$ pair, we plot the contours of $\mathcal{L}_s - \mathcal{L}_t$ for a given $\gamma_s, \gamma_t, \sigma_\varepsilon$. In these plots, the solid red curves show the pairs $(\lambda_w, \lambda_s)$ for which $\mathcal{L}_s = \mathcal{L}_t$. The left plot corresponds to the case where $\gamma_s < 1$. It can be seen that when $\lambda_t < \lambda_t^\star$, the student model outperforms the teacher as long $\lambda_s < \bar{\lambda}(\lambda_t; \gamma_s, \gamma_t, \sigma_\varepsilon)$. Moreover, the student is always worse than the teacher when $\lambda_t > \lambda_t^\star$. The right plot corresponds to the case with $\gamma_s > 1$. In this case, it is seen that as predicted in Theorem 5, for some values of $\lambda_t$, the student outperforms the teacher only if $\lambda_s \in (\bar{\lambda}_-, \bar{\lambda}_+)$ for some $0 < \bar{\lambda}_- < \bar{\lambda}_+$. See Section 4 for more details on the experimental setting.

**Mechanism of Weak-to-Strong Generalization.**    In this section, we show that the student's reduced error stems from *compensating for the teacher's insufficient regularization*. Thus, intuitively, similar to what is proven in Theorem 5, when the teacher is already over-regularized, the student is unable to achieve a better performance by leveraging this mechanism. Also, note that when $\gamma_s > 1$, the student model is over-parameterized and as a result, some information is lost. Thus, more regularization is required for the student to outperform the teacher. This can be seen as the reason why in this regime, a non-zero lower bound exists for $\lambda_s$ to ensure this. Our results complement the results of Medvedev et al. [2025] who demonstrated that regularizing the student is essential to avoid overfitting to the mistakes in a setting where the teacher is optimally trained.

**Non-Monotone Student Test Error Curves.**    Studying the test error $\mathcal{L}_s$ as a function of the the teacher overparameterization $\gamma_t$ using Theorem 4, we observe that student model also exhibits the *double descent* phenomenon, where we can see a second bias-variance tradeoff in the test error beyond the interpolation limit [Belkin et al., 2019]. See Section E for more details. This is in line with findings in different linear regression setups such as standard ridge/ridgeless regression [Hastie et al., 2022, Nakkiran et al., 2021] (which corresponds to double descent in the teacher model in our setting), ridge regression with correlated samples [Atanasov et al., 2024, Moniri and Hassani, 2025], and weighted ridge regression [Wu and Xu, 2020].

**Universality.**    Similar to other high-dimensional ridge regression problems (see e.g., Hastie et al. [2022], Hu and Lu [2023], Montanari and Saeed [2022], etc.), we can use a simple Lindeberg exchange argument [Lindeberg, 1922, Korada and Montanari, 2011] to show that $\Delta$ from Theorem 4 exhibits universality; i.e., the actual distribution of the samples does not matter and the limiting value of $\mathcal{L}_s - \mathcal{L}_t$ is determined only by the first and second moment of the covariates. Thus, our results in theory do hold much more broadly than the Gaussian case. The study of this universality is not central to our discussion and we leave it as future work.

**Other Related Work.**    The high-dimensional ridge regression setting considered in this section is related to the linear regression setting considered by Dohmatob et al. [2024] to study model collapse. However, in their setting, only the downstream model (which corresponds to the student model in our setting) has a non-zero ridge regularization. Similar settings have also been studied in the self-distillation literature (see e.g., Das and Sanghavi [2023], Pareek et al. [2024], etc.). However, the training procedure of the models are different; e.g. in their setting, the teacher generates synthetic labels for its own training set and not for a fresh set of covariates. Additionally, in the self distillation setting, the student model still has access to ground truth labels.

## 2.2   Setting 2: High-Dimensional Weighted Ridge Regression

In this section, we consider a setting where the strong model is a linear model trained using weighted ridge regression [Hoerl and Kennard, 1970, Casella, 1980, Wu and Xu, 2020, Richards et al., 2021]. Given samples $\mathcal{S} \subset \mathbb{R}^{d_x} \times \mathbb{R}$, the estimator $\mathrm{WRidge}(\mathcal{S}, \lambda, \mathbf{\Gamma})$ is defined as

$$\mathrm{WRidge}(\mathcal{S}, \lambda, \mathbf{\Gamma}) := \underset{\boldsymbol{\beta} \in \mathbb{R}^{d_x}}{\mathrm{argmin}} \left[ \frac{1}{n} \sum_{(\mathbf{x}_i, y_i) \in \mathcal{S}} \left( y_i - \boldsymbol{\beta}^\top \mathbf{x}_i \right)^2 + \lambda \|\mathbf{\Gamma}^{-1} \boldsymbol{\beta}\|_2^2 \right], \qquad (6)$$

where $\mathbf{\Gamma} \in \mathbb{R}^{d_x \times d_x}$ is a weighting matrix and $\lambda \in \mathbb{R}$ is a scalar. In this section, we assume that the teacher is still a (standard) ridge regression estimator. However, unlike the previous section, we let the student model be a weighted ridge estimator; i.e.,

$$\hat{\boldsymbol{\beta}}_t = \mathrm{WRidge}(\mathcal{S}_t, \lambda_t, \mathbf{I}_{d_x}), \quad \hat{\boldsymbol{\beta}}_s = \mathrm{WRidge}(\mathcal{S}_s, \lambda_s, \mathbf{\Gamma}). \qquad (7)$$

In this model, the matrix $\mathbf{\Gamma}$ is assumed to be given and fixed. The matrix $\mathbf{\Gamma}$ determines the structure of the student regularization enforcing different levels of regularization in different directions. In the next remark, we provide a linear neural-network interpretation for $\mathbf{\Gamma}$.

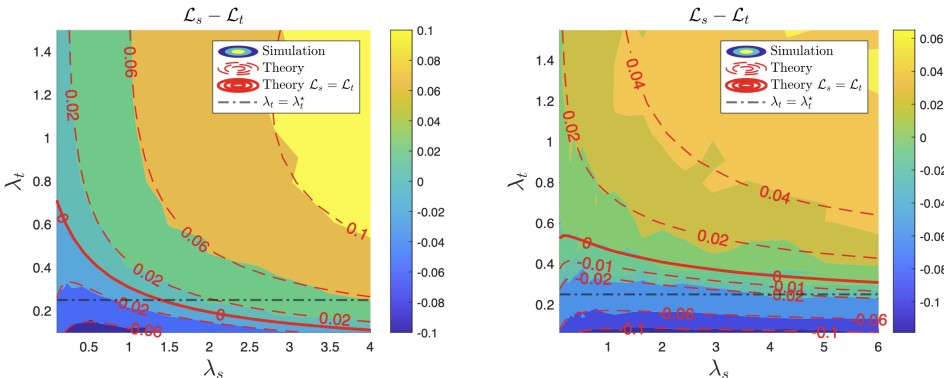

Figure 2: Test-error difference $\mathcal{L}_s - \mathcal{L}_t$ as a function of $(\lambda_t, \lambda_s)$ in the setting of Section 2.2. Filled contours are numerical simulations; dashed red contours follow the theory of Theorem 8. The solid curve marks $\mathcal{L}_s = \mathcal{L}_t$, and the dashed black curve is $\lambda_t = \lambda_t^\star$. **Left**: under-parameterized student. **Right**: over-parameterized student. See Section 4 for more details.

**Remark 6.** *The weighted ridge estimator* $\mathrm{WRidge}(\mathcal{S}, \lambda, \boldsymbol{\Gamma})$ *can also be seen as training the second layer of a two-layer linear neural network* $f_{\mathrm{NN}}(\mathbf{x}) = \mathbf{x}^\top \boldsymbol{\Gamma} \boldsymbol{\alpha}$*; i.e.,* $\hat{\boldsymbol{\beta}} = \boldsymbol{\Gamma} \hat{\boldsymbol{\alpha}}$ *in which*

$$\hat{\boldsymbol{\alpha}} = \underset{\boldsymbol{\alpha} \in \mathbb{R}^{d_{\mathsf{x}}}}{\mathrm{argmin}} \left[ \frac{1}{n} \sum_{(\mathbf{x}_i, y_i) \in \mathcal{S}} \left( y_i - \mathbf{x}_i^\top \boldsymbol{\Gamma} \boldsymbol{\alpha} \right)^2 + \lambda \|\boldsymbol{\alpha}\|_2^2 \right].$$

In light of the connection to linear neural networks in Remark 6, one can think of the student model as a pre-trained neural network. During the pre-training, we assume that the student model has had access to data from various sources with a shared structure with $\boldsymbol{\beta}_\star$. The goal of the pre-training is to use this data to learn features that align well with the underlying task $\boldsymbol{\beta}_\star$ [Sun et al., 2021]. Motivated by a recent line of results in deep learning theory where the updated first layer weights are shown to have a *spiked structure* with a few directions having information about the target function [Ba et al., 2022, Moniri et al., 2024, Cui et al., 2024, Zhang et al., 2025, Ba et al., 2024, Demir and Doğan, 2024, Moniri and Hassani, 2024, Li and Sonthalia, 2024, Mousavi-Hosseini et al., 2023, Radhakrishnan et al., 2024], we model the alignment of $\boldsymbol{\Gamma}$ with the task structure using a non-informative bulk component plus an informative low-rank component.

**Assumption 7.** *We assume that the matrix* $\boldsymbol{\Gamma} \in \mathbb{R}^{d_{\mathsf{x}} \times d_{\mathsf{x}}}$ *is given by*

$$\boldsymbol{\Gamma} = \mathbf{I}_{d_{\mathsf{x}}} + d_{\mathsf{x}} \hat{\boldsymbol{\beta}} \hat{\boldsymbol{\beta}}^\top \quad \textit{with} \quad \frac{|\hat{\boldsymbol{\beta}}^\top \boldsymbol{\beta}_\star|}{\|\hat{\boldsymbol{\beta}}\|_2 \|\boldsymbol{\beta}_\star\|_2} \to_{\mathbb{P}} \zeta \tag{8}$$

*where* $\zeta \in [0, 1]$ *is the correlation of the learned direction* $\hat{\boldsymbol{\beta}}$ *to the target direction* $\boldsymbol{\beta}_\star$ *which is a measure of how much* $\hat{\boldsymbol{\beta}}$ *aligns with the target direction* $\boldsymbol{\beta}_\star$.

The prefactor $d_{\mathsf{x}}$ for the spike term in (8) is chosen in a way to ensure that $\|\mathbf{I}\|_{\mathrm{Fr}} \asymp \|d_{\mathsf{x}} \hat{\boldsymbol{\beta}} \hat{\boldsymbol{\beta}}^\top\|_{\mathrm{Fr}}$. This closely resembles the scaling of the updated weights with maximal update parameterization in the feature learning theory literature [Yang and Hu, 2021, Ba et al., 2022]. In the following theorem, we characterize the test error difference of the models $\mathcal{L}_s - \mathcal{L}_t$ for this setting in the high-dimensional proportional regime of Assumption 1.

**Theorem 8.** *Under the conditions of Proposition 3, the test errors of the student and teacher model from (7) with* $\boldsymbol{\Gamma}$ *from Assumption 7 satisfy* $\mathcal{L}_s - \mathcal{L}_t \to_{\mathbb{P}} \Delta - \zeta^2 \Delta_{\boldsymbol{\Gamma}}$ *where the expression for* $\Delta$ *is given in Theorem 4, and*

$$\Delta_{\boldsymbol{\Gamma}} := \lambda_s \left( -1 + \lambda_t m_{t,1} \right) \left[ -2 \lambda_t m_{s,1} m_{t,1} + \lambda_s m_{s,2} (-1 + \lambda_t m_{t,1}) \right].$$

*where* $m_{t,1}, m_{t,2}, m_{s,1}, m_{s,2}$ *are defined in Definition 2. Additionally, we have* $\Delta_{\boldsymbol{\Gamma}} \geq 0$.

In the limiting formula for $\mathcal{L}_s - \mathcal{L}_t$, the term $\Delta$ is equal to the limiting value $\mathcal{L}_s - \mathcal{L}_t$ in the setting of Section 2.1 where both models are trained using (standard) ridge regression, and the benefit of the learned features for the student is due to the term $-\zeta^2 \Delta_{\Gamma}$, which is always non-positive. Because of this term, even in the settings that $\Delta \geq 0$ (i.e., the mechanism of Section 2.1 is not enough on its own for weak-to-strong generalization), the student may still outperform the teacher.

Figures 2 and 3 demonstrate that the asymptotic characterization of Theorem 8 match simulations very well even for moderately large $n_s, n_t, d_x$. In Figure 2, we fix $\gamma_s, \gamma_t$ and $\sigma_\varepsilon^2$ and plot the contours of $\mathcal{L}_s - \mathcal{L}_t$ for different $(\lambda_t, \lambda_s)$ pairs. We show that unlike Section 2.1, in both settings with $\gamma_s > 1$ or $\gamma_s < 1$, pairs $(\lambda_t, \lambda_s)$ with $\lambda_t > \lambda_t^\star = \sigma_\varepsilon^2 \gamma_t$ (i.e., over-regularized teacher) exist where the student model outperforms the teacher. In Figure 3, we set $\lambda_t = \lambda_t^\star$ and plot $\mathcal{L}_s$ as a function of $\lambda_s$ for different values of the feature quality parameter $\zeta$. We see that for small $\zeta$, the student never outperforms the teacher, similar to the case in Section 2.1. However, this changes when $\zeta$ is increased, and the student can have a smaller test error for some values of $\lambda_s$.

**Mechanism of Weak-to-Strong Generalization.** Theorem 8 shows that if the student model has been pre-trained and has learned features that are better suited for the task of predicting the target function (or equivalently is fine-tuned using a better regularization structure), it can leverage this advantage to achieve a better performance compared to the teacher, despite being trained on labels generated by the teacher. This shows yet another mechanism of weak-to-strong generalization.

## 3 The Nonlinear Case

In Section 2, we considered a setting where both the student and the teacher model trained a linear head through a convex objective. Although this is a very effective model for the analysis of the roles of regularization and over-parameterization, it coincides to the linearized regime of neural network training where features are frozen at their initialization. As a result, the linearized models are not rich enough to study phenomena that happen as a result of (nonlinear) *feature learning*.

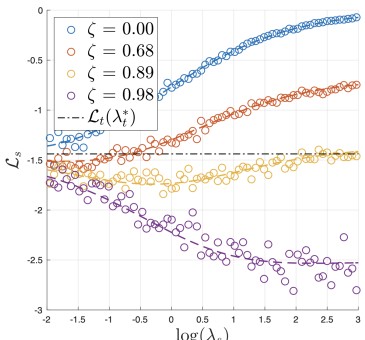

Consider the problem of learning a few distinct skills and abilities using data from a compositional task, where a blend of different skills are needed in order to succeed. We model each skill as a vector in the high-dimensional input space $\mathbb{R}^{d_x}$, and learning a skill as learning the corresponding direction. Take, as a running example the task of holding a coherent conversation in an unfamiliar language. To succeed, the model needs to blend abilities such as logical reasoning, with language-specific skills. Abilities such as logical reasoning are difficult to acquire, but transferable across domains. However, language-specific abilities such as vocabulary and grammar, are conceptually straightforward, but they are task-specific and often cannot be learned from other related tasks.

Figure 3: Student error $\mathcal{L}_s$ versus $\log \lambda_s$ in the setting of Section 2.2, plotted for several values of $\zeta$ with the teacher optimally regularized. Circles show simulation results, and dashed curves are the predictions of Theorem 8. The dashed black line marks the teacher error $\mathcal{L}_t$. See Section 4 for details.

Motivated by this setting, in our model, we let the samples for the teacher model be generated according to a multi-index function (e.g., Box and Cox [1964], Bickel and Doksum [1981]) with an easy and a hard component. Here, the teacher has access to $\mathcal{S}_t = \{(\mathbf{x}_i, y_i)\}_{i=1}^{n_t}$ drawn from

$$\mathbf{x}_i \sim \mathsf{N}(\mathbf{0}, \mathbf{I}_{d_x}), \quad \text{and} \quad y_i = \sigma_e(\mathbf{x}^\top \boldsymbol{\beta}_e) + \sigma_h(\mathbf{x}^\top \boldsymbol{\beta}_h), \tag{9}$$

where $\boldsymbol{\beta}_e, \boldsymbol{\beta}_h \in \mathbb{R}^{d_x}$ are two orthonormal directions that we want to learn, and $\sigma_e, \sigma_h : \mathbb{R} \to \mathbb{R}$ are two link functions. In this problem, the hardness of learning each direction $\boldsymbol{\beta}_e, \boldsymbol{\beta}_h$ from these samples is known to be characterized by the *information-exponent* of their link function [Dudeja and Hsu, 2018, Ben Arous et al., 2021]. For any real-valued function $\sigma : \mathbb{R} \to \mathbb{R}$ with Hermite coefficients $\{c_{\sigma,k}\}_{k=0}^\infty$, the information-exponent is defined as $\kappa_\sigma := \min\{k \in \mathbb{N} : c_{\sigma,k} \neq 0\}$. We make the following assumption on $\sigma_e$ and $\sigma_h$.

**Assumption 9.** *Assume that $\kappa_{\sigma_e} = 1$ and $\kappa_{\sigma_h} > 1$ (i.e., $\sigma_e$ is an easy and $\sigma_h$ is a hard link function).*

We let the student $\hat{f}_s$ and teacher $\hat{f}_s$ models be neural networks given by $\hat{f}_s(\mathbf{x}) = \mathbf{a}_s^\top \sigma(\mathbf{W}_s \mathbf{x})$ and $\hat{f}_t(\mathbf{x}) = \mathbf{a}_t^\top \sigma(\mathbf{W}_t \mathbf{x})$, where $\sigma : \mathbb{R} \to \mathbb{R}$ is an activation function with $\kappa_\sigma = 1$, and $\mathbf{a}_t \in \mathbb{R}^{p_t}, \mathbf{a}_s \in \mathbb{R}^{p_s}$, $\mathbf{W}_t \in \mathbb{R}^{p_t \times d_\mathsf{x}}$, and $\mathbf{W}_s \in \mathbb{R}^{p_s \times d_\mathsf{x}}$. To learn the relevant directions $\boldsymbol{\beta}_e, \boldsymbol{\beta}_h$, we update the first layer weights to align them to these directions; a task also referred to as *weak recovery* [Ben Arous et al., 2021, Dandi et al., 2023, 2024, Arnaboldi et al., 2024, Lee et al., 2024].

For training, we update the teacher model $\hat{f}_t$ using the samples $\mathcal{S}$ and use $\hat{f}_t$ to generate synthetic labels for $n_s \in \mathbb{N}$ unlabeled covariates $\mathcal{S}_s = \{\tilde{\mathbf{x}}_i\}_{i=1}^{n_s}$ drawn from the same distribution according to $\tilde{\mathbf{x}}_i \sim \mathsf{N}(\mathbf{0}, \mathbf{I}_{d_\mathsf{x}})$ as $\tilde{y}_i = \hat{f}_t(\tilde{\mathbf{x}}_i)$. We then use these samples to update the student model $\hat{f}_s$. We consider the correlation loss defined as

$$\widehat{\mathcal{L}}_t := -n_t^{-1} \sum_{i=1}^{n_t} y_i \hat{f}_t(\mathbf{x}_i), \quad \text{and} \quad \widehat{\mathcal{L}}_s := -n_s^{-1} \sum_{i=1}^{n_s} \tilde{y}_i \hat{f}_s(\tilde{\mathbf{x}}_i). \tag{10}$$

Fixing $\mathbf{a}_t \in \mathbb{R}^{p_t}$ with $\|\mathbf{a}_t\|_2 = \Theta(1)$, and initializing $\mathbf{W}_t$ at $\mathbf{W}_{t,0}$ with i.i.d. $\mathsf{N}(0, d_\mathsf{x}^{-1})$ entries, we update $\mathbf{W}_t$ using one-step of gradient descent on $\mathcal{L}_t$ given by

$$\widehat{\mathbf{W}}_t = \mathbf{W}_{t,0} - \eta_t \nabla_{\mathbf{W}_t} \widehat{\mathcal{L}}_t |_{\mathbf{W}_{t,0}, \mathbf{a}_t}.$$

The following result, which is a corollary of Ba et al. [2022, Proposition 2], shows that the after this update, $\widehat{\mathbf{W}}_t$ aligns to the easy direction $\boldsymbol{\beta}_e$, but does not align to the hard direction $\boldsymbol{\beta}_h$; i.e., the teacher model *could not learn the hard direction* using these samples.

**Proposition 10.** *Under the high-dimensional proportional limit of Assumption 1, and assuming that $\eta_t = O(1)$ and $p_t = \Theta(d_\mathsf{x})$, we have*

$$\|\widehat{\mathbf{W}}_t \boldsymbol{\beta}_e\|_2 \to_{\mathbb{P}} c > 0 \quad \text{and} \quad \|\widehat{\mathbf{W}}_t \boldsymbol{\beta}_h\|_2 \to_{\mathbb{P}} 0.$$

After this update, we set $\hat{f}_t(\mathbf{x}) = \mathbf{a}_t^\top \sigma(\hat{\mathbf{W}}_t \mathbf{x})$. We assumed that the teacher has not gone through extensive pre-training on other tasks that depend on the directions $\boldsymbol{\beta}_e, \boldsymbol{\beta}_h$; thus, we made the assumption that the $\mathbf{W}_t$ is initialized at random. Recall that $\hat{\boldsymbol{\beta}}_h$ is a hard direction, but it is relevant for a variety of tasks. Thus, we assume that the student has been pre-trained on a variety of different relevant tasks, and has already learned the hard but cross-domain ability that corresponds to $\boldsymbol{\beta}_h$; thus $\mathbf{W}_s$ at initialization is aligned to $\boldsymbol{\beta}_h$. In particular, we set $\mathbf{W}_{s,0} = \bar{\mathbf{W}}_{s,0} + \tau \bar{\mathbf{a}} \boldsymbol{\beta}_h^\top$ where $\bar{\mathbf{W}}_{s,0} \in \mathbb{R}^{p_s \times d_\mathsf{x}}$ has $\mathsf{N}(0, d_\mathsf{x}^{-1})$ entries, $\tau \in \mathbb{R}$ and $\bar{\mathbf{a}} \in \mathbb{R}^{p_s}$ is a unit norm vector. We update $\mathbf{W}_s$ using one-step of gradient descent on $\mathcal{L}_s$; i.e.,

$$\widehat{\mathbf{W}}_s = \mathbf{W}_{s,0} - \eta_s \nabla_{\mathbf{W}_s} \widehat{\mathcal{L}}_s |_{\mathbf{W}_{s,0}, \mathbf{a}_s}.$$

Note that training $\mathbf{W}_s$ excessively on data generated from the teacher can result in the student to forget the direction $\boldsymbol{\beta}_h$. However, in the next theorem, we show that a single step of SGD on $\widehat{\mathcal{L}}_s$ can induce non-trivial alignment between $\mathbf{W}_s$ and the easy direction $\boldsymbol{\beta}_e$ while keeping the weights aligned to $\boldsymbol{\beta}_h$. This aligns to the empirical and theoretical findings of Burns et al. [2024], Medvedev et al. [2025] that show that early stopping the teacher is required for weak-to-strong generalization.

**Theorem 11.** *In the asymptotic regime of Assumption 1 with $p_t, p_s = \Theta(d_\mathsf{x})$, assuming that $\eta_t, \eta_s = O(1)$ and $\tau = o(\sqrt{d_\mathsf{x}})$, we have*

$$\|\widehat{\mathbf{W}}_s \boldsymbol{\beta}_e\|_2 \to_{\mathbb{P}} c_e > 0 \quad \text{and} \quad \|\widehat{\mathbf{W}}_s \boldsymbol{\beta}_h\|_{\mathrm{op}} \to_{\mathbb{P}} c_h > 0$$

This theorem is a extension of [Ba et al., 2022, Proposition 2] to the case where the first-layer weight is initialized as a spiked random matrix and the labels are also generated by another one-step updated two-layer neural network, which can be also of independent interest. This theorem shows that, under this setting, the student model is still able to learn the direction $\boldsymbol{\beta}_e$ from the imperfect labels generated by the teacher, and achieve non-vanishing alignment to both the directions $\boldsymbol{\beta}_e, \boldsymbol{\beta}_h$, although the student was unable to learn the hard direction $\boldsymbol{\beta}_h$.

**Mechanism of Weak-to-Strong Generalization.** In this setting, a teacher model can acquire the easier, yet specialized skills through fine-tuning, even though it cannot learn abilities that are challenging to learn. By contrast, the student is pretrained on vast, heterogeneous corpora and already possesses those hard, yet cross-domain, skills. As a result, weak-to-strong generalization can happen when the teacher teaching the student the specialized language abilities, thereby complementing the student's strengths from pre-training.

# 4 Numerical Validation

In this section, we provide the details of the simulations presented throughout the papers.

**Figures 1 and 2.** We fix the values of $d_x, n_t, n_s$ and $\sigma_\varepsilon$ and plot the contours of $\mathcal{L}_s - \mathcal{L}_t$ using numerical simulations and also the results of Theorem 4 and 8. The simulation results are averaged over ten trials. See Section 2.1 and 2.2 for discussions of the results. In Figure 1, for the under-parameterized regime (**left**), we set $d_x = 500, n_t = n_s = 2000, \sigma_\varepsilon = 1$ and for the over-parameterized regime (**right**), we set $d_x = 500, n_t = 2000, n_s = 416, \sigma_\varepsilon = 2$. In Figure 2, for the under-parameterized regime (**left**), we set $\zeta = 0.8, d_x = 500, n_t = n_s = 2000, \sigma_\varepsilon = 1$, and the over-parameterized regime (**right**), we set $\zeta = 0.88, d_x = 500, n_t = 2000, n_s = 416, \sigma_\varepsilon = 1$.

**Figure 3.** In these experiments, we set $d_x = 500, n_t = n_s = 2000, \sigma_\varepsilon = 1$ and set $\lambda_t = \sigma_\varepsilon^2 \gamma_t = 0.25$. We compare the theoretical curves of $\mathcal{L}_s$ as a function of $\gamma_s$ with numerical simulation, for $\zeta \in \{0, 0.68, 0.89, 0.98\}$. The simulations in this experiment have not been averaged over multiple trials. See Section 2.2 for discussion of the results.

# 5 Conclusion

In this paper, we show that weak-to-strong generalization is not unique to complex language models and can happen even in much simpler theoretical setups. By studying three natural and tractable learning theoretical settings, we identified and theoretically analyzed three distinct routes by which a student model can outperform its teacher: (1) *compensating for under-regularization*, (2) *harnessing a more task-aligned regularization structure*, and (3) *combining teacher-taught, easy-to-learn components with pretrained, hard-to-learn features*. Our results clarify when and why these effects arise, complementing prior empirical and theoretical insights about the roles of regularization and overparameterization, and the role of feature adaptation.

# 6 Acknowledgments

Behrad Moniri gratefully acknowledges the gift from AWS AI to Penn Engineering's ASSET Center for Trustworthy AI. The work of Behrad Moniri and Hamed Hassani is supported by The Institute for Learning-enabled Optimization at Scale (TILOS), under award number NSF-CCF-2112665, and the NSF CAREER award CIF-1943064.

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

# A Preliminaries

Given two matrices $\mathbf{A}, \mathbf{B} \in \mathbb{R}^{n_1 \times n_2}$, we denote their Hadamard product (element-wise product) by $\mathbf{A} \odot \mathbf{B} \in \mathbb{R}^{n_1 \times n_2}$. Also, for for $k \in \mathbb{N}$, we define the Hadamard power as

$$\mathbf{A}^{\odot k} = \underbrace{\mathbf{A} \odot \cdots \odot \mathbf{A}}_{k \text{ times}} \in \mathbb{R}^{n_1 \times n_2}.$$

**Lemma 12.** *For any* $\mathbf{v} \in \mathbb{R}^{n_1}$, $\mathbf{u} \in \mathbb{R}^{n_2}$, *and* $\mathbf{C} \in \mathbb{R}^{n_1 \times n_2}$, *we have*

$$(\mathbf{v}\mathbf{u}^\top) \odot \mathbf{C} = \operatorname{diag}(\mathbf{v})\,\mathbf{C}\,\operatorname{diag}(\mathbf{u}).$$

*Proof.* The proof is immediate by writing the entries of the two sides. $\qquad\square$

We also heavily leverage the following theorem to prove the concentration inequality for quadratic forms. See e.g., Rudelson and Vershynin [2013] for a modern proof.

**Theorem 13** (Hanson-Wright Inequality [Hanson and Wright, 1971]). *Let* $\mathbf{x} = (X_1, \ldots, X_n) \in \mathbb{R}^d$ *be a random vector with independent sub-gaussian components* $X_i$ *with* $\mathsf{E}X_i = 0$. *Let* $\mathbf{D}$ *be an* $n \times n$ *matrix. Then, for every* $t \geq 0$, *we have*

$$\mathbb{P}\Big[\,\big|\mathbf{x}^\top \mathbf{D}\,\mathbf{x} - \mathsf{E}\big[\mathbf{x}^\top \mathbf{D}\,\mathbf{x}\big]\big| > t\Big] \leq 2\exp\left[-c\min\left(\frac{t^2}{\|\mathbf{D}\|_F^2}, \frac{t}{\|\mathbf{D}\|_{\mathrm{op}}}\right)\right],$$

*where* $c$ *is a constant that depends only on the sub-gaussian constants of* $X_i$.

To analyze spiked random matrices, we will use the following matrix identity.

**Lemma 14.** *(Sherman-Morrison Formula). Let* $\mathbf{A} \in \mathbb{R}^{n \times n}$ *be an invertible matrix, and let* $\mathbf{u}, \mathbf{v} \in \mathbb{R}^n$ *be column vectors such that* $1 + \mathbf{v}^\top \mathbf{A}^{-1}\mathbf{u} \neq 0$. *Then the inverse of the rank-one update* $\mathbf{A} + \mathbf{u}\mathbf{v}^\top$ *is given by:*

$$(\mathbf{A} + \mathbf{u}\mathbf{v}^\top)^{-1} = \mathbf{A}^{-1} - \frac{\mathbf{A}^{-1}\mathbf{u}\mathbf{v}^\top \mathbf{A}^{-1}}{1 + \mathbf{v}^\top \mathbf{A}^{-1}\mathbf{u}}.$$

## A.1 Hermite Polynomials

We let $H_k$ be the $k$-th (probabilist's) Hermite polynomial on $\mathbb{R}$ defined by

$$H_k(x) = (-1)^k \exp(x^2/2)\frac{d^k}{dx^k}\exp(-x^2/2) \quad \forall x \in \mathbb{R}.$$

These polynomials form an orthogonal basis in the Hilbert space $L^2$ of measurable functions $f : \mathbb{R} \to \mathbb{R}$ such that

$$\int f^2(x)e^{-\frac{x^2}{2}}\,dx < \infty$$

with inner product

$$\langle f, g \rangle = \int f(x)g(x)\,e^{-\frac{x^2}{2}}\,dx.$$

The first few Hermite polynomials are

$$H_0(x) = 1, \quad H_1(x) = x, \quad \text{and} \quad H_2(x) = x^2 - 1.$$

**Lemma 15.** *For any* $k \in \mathbb{N}$ *and* $x, y \in \mathbb{R}$, *we have*

$$H_k(x + y) = \sum_{j=0}^{k} \binom{k}{j} x^j H_{k-j}(y)$$

*Proof.* Note that using [Abramowitz and Stegun, 1968, Equation 22.8.8] we have

$$\frac{d}{dx}H_k(x) = kH_{k-1}(x).$$

Thus, the $j$-th derivative of $H_k$ is given by

$$\frac{d^j}{dx^j} H_k(x) = \frac{k!}{(k-j)!} H_{k-j}(x).$$

By Taylor expanding $H_k(x+y)$ at $y$, we find

$$H_k(x+y) = \sum_{j=0}^{k} \frac{x^j}{j!} \frac{d^j}{dy^j} H_k(y) = \sum_{j=0}^{k} \binom{k}{j} x^j H_{k-j}(y),$$

proving the lemma.

## A.2 Random Matrix Theory

We first define the following empirical covariance and resolvent matrices that will appear throughout the proofs.

$$\hat{\mathbf{\Sigma}} = \mathbf{X}^\top \mathbf{X} / n_t, \quad \tilde{\mathbf{\Sigma}} = \tilde{\mathbf{X}}^\top \tilde{\mathbf{X}} / n_s,$$
$$\hat{\mathbf{R}} = (\hat{\mathbf{\Sigma}} + \lambda_t \mathbf{I}_{\mathrm{dx}})^{-1}, \quad \text{and} \quad \tilde{\mathbf{R}} = (\tilde{\mathbf{\Sigma}} + \lambda_s \mathbf{I}_{\mathrm{dx}})^{-1}.$$

We will use the following characterization of the eigenvalues of $\hat{\mathbf{\Sigma}}, \tilde{\mathbf{\Sigma}}$ in the high-dimensional proportional limit by Marchenko and Pastur [1967].

**Theorem 16** (Marchenko–Pastur Theorem)**.** *In the high-dimensional proportional limit where $n_t, n_s, \mathrm{d_x} \to \infty$ such that $\frac{\mathrm{dx}}{n_s} \to \gamma_s$ and $\frac{\mathrm{dx}}{n_t} \to \gamma_t$, the empirical spectral distribution (ESD) of $\hat{\mathbf{\Sigma}}$ (and $\tilde{\mathbf{\Sigma}}$) converges almost surely to the Marchenko–Pastur distribution $\mu_{MP(\gamma_t)}$ (and $\mu_{MP(\gamma_s)}$).*

Recall from definition 2 that the function $m(\cdot, \cdot) \to \mathbb{R}$ is defined as

$$m(\lambda; \gamma) = \int \frac{\mathrm{d}\mu_{MP(\gamma)}(s)}{s + \lambda}.$$

Hence, taking derivatives with respect to $\lambda$, we get

$$m'(\lambda; \gamma) = \frac{\partial}{\partial \lambda} m(\lambda; \gamma) = -\int \frac{\mathrm{d}\mu_{MP(\gamma)}(s)}{(s + \lambda)^2}.$$

In the following section, we write the test error $\mathcal{L}_s$ and $\mathcal{L}_t$ as a function of $m$ and its derivatives. In particular, note that using the Marchenko-Pastur theorem, we have

$$\mathrm{d_x^{-1}} \operatorname{Tr}\left(\hat{\mathbf{R}}\right) \to_{\mathbb{P}} m_{t,1}, \quad \mathrm{d_x^{-1}} \operatorname{Tr}\left(\hat{\mathbf{R}}^2\right) \to_{\mathbb{P}} m_{t,2}, \quad \text{and}$$

$$\mathrm{d_x^{-1}} \operatorname{Tr}\left(\tilde{\mathbf{R}}\right) \to_{\mathbb{P}} m_{s,1}, \quad \mathrm{d_x^{-1}} \operatorname{Tr}\left(\tilde{\mathbf{R}}^2\right) \to_{\mathbb{P}} m_{s,2},$$

where for $p \in \{s, t\}$, we define $m_{p,1} = m(\lambda_p, \gamma_p)$ and $m_{p,2} = -\frac{\partial m}{\partial \lambda}\big|_{\lambda_p, \gamma_p}$.

In the following proofs, we will also use the asymptotic freeness of independent Wishart random matrices [Voiculescu, 1991, Capitaine and Donati-Martin, 2007]. See [Feier, 2012, Section 3.4 and 4.4] for a brief overview of free probability theory for random matrix theory.

## B Proof of Proposition 3

To prove this theorem, note that the vector $\hat{\boldsymbol{\beta}}_t$ can be written as

$$\hat{\boldsymbol{\beta}}_t = (\mathbf{X}^\top \mathbf{X} + \lambda_t n_t \mathbf{I}_{\mathrm{dx}})^{-1} \mathbf{X}^\top \mathbf{y}$$

$$= (\mathbf{X}^\top \mathbf{X} + \lambda_t n_t \mathbf{I}_{\mathrm{dx}})^{-1} \mathbf{X}^\top (\mathbf{X} \boldsymbol{\beta}_\star + \boldsymbol{\varepsilon})$$

$$= (\hat{\mathbf{\Sigma}} + \lambda_t \mathbf{I}_{\mathrm{dx}})^{-1} \hat{\mathbf{\Sigma}} \boldsymbol{\beta}_\star + n_t^{-1} (\hat{\mathbf{\Sigma}} + \lambda_t \mathbf{I}_{\mathrm{dx}})^{-1} \mathbf{X}^\top \boldsymbol{\varepsilon},$$

where we have used the fact that $\mathbf{y} = \mathbf{X}\boldsymbol{\beta}_\star + \boldsymbol{\varepsilon}$. Thus, recalling the definition of $\hat{\boldsymbol{\Sigma}}$ and $\hat{\mathbf{R}}$ from the Section A, we can write

$$\hat{\boldsymbol{\beta}}_t - \boldsymbol{\beta}_\star = (\hat{\mathbf{R}}\,\hat{\boldsymbol{\Sigma}} - \mathbf{I}_{\mathrm{dx}})\,\boldsymbol{\beta}_\star + n_t^{-1}\hat{\mathbf{R}}\,\mathbf{X}^\top\boldsymbol{\varepsilon}$$

The test error of the teacher model is given by $\mathcal{L}_t = \sigma_\varepsilon^2 + \|\hat{\boldsymbol{\beta}} - \boldsymbol{\beta}_\star\|_2^2$ in which

$$\|\hat{\boldsymbol{\beta}}_t - \boldsymbol{\beta}_\star\|_2^2 = \boldsymbol{\beta}_\star^\top(\hat{\mathbf{R}}\,\hat{\boldsymbol{\Sigma}} - \mathbf{I}_{\mathrm{dx}})^\top(\hat{\mathbf{R}}\,\hat{\boldsymbol{\Sigma}} - \mathbf{I}_{\mathrm{dx}})\boldsymbol{\beta}_\star + n_t^{-2}\boldsymbol{\varepsilon}^\top\left(\mathbf{X}\,\hat{\mathbf{R}}^2\mathbf{X}^\top\right)\boldsymbol{\varepsilon} + o_\mathbb{P}(1),$$

where we have used the Hanson-Wright inequality and the facts that $\boldsymbol{\varepsilon}$ and $\boldsymbol{\beta}_\star$ are independent mean zero random vectors. Again, by using the Hanson-Wright inequality and recalling that $(\boldsymbol{\varepsilon}, \boldsymbol{\beta}_\star)$ is independent of other sources of randomness in the problem, we can further simplify the above expression to arrive at

$$\|\hat{\boldsymbol{\beta}}_t - \boldsymbol{\beta}_\star\|_2^2 \to_\mathbb{P} \mathrm{d}_\mathsf{x}^{-1}\,\mathrm{Tr}\left[(\hat{\mathbf{R}}\,\hat{\boldsymbol{\Sigma}} - \mathbf{I}_{\mathrm{dx}})^\top(\hat{\mathbf{R}}\,\hat{\boldsymbol{\Sigma}} - \mathbf{I}_{\mathrm{dx}})\right] + \sigma_\varepsilon^2\gamma_t\mathrm{d}_\mathsf{x}^{-1}\,\mathrm{Tr}\left[\hat{\boldsymbol{\Sigma}}\,\hat{\mathbf{R}}^2\right]$$

Note that $\hat{\mathbf{R}}\hat{\boldsymbol{\Sigma}} = \mathbf{I}_{\mathrm{dx}} - \lambda_t\hat{\mathbf{R}}$. Thus, denoting the eigenvalues of $\hat{\boldsymbol{\Sigma}}$ by $\{\sigma_k\}_{k=1}^d$, we can use the Marchenko-Pastur Theorem for covariance matrices to write

$$\mathrm{d}_\mathsf{x}^{-1}\,\mathrm{Tr}\left[(\hat{\mathbf{R}}\,\hat{\boldsymbol{\Sigma}} - \mathbf{I}_{\mathrm{dx}})^\top(\hat{\mathbf{R}}\,\hat{\boldsymbol{\Sigma}} - \mathbf{I}_{\mathrm{dx}})\right] = \lambda_t^2\mathrm{d}_\mathsf{x}^{-1}\,\mathrm{Tr}\left[\hat{\mathbf{R}}^2\right] \to_\mathbb{P} \lambda_t^2 m_{t,2}.$$

Similarly, we have

$$\mathrm{d}_\mathsf{x}^{-1}\,\mathrm{Tr}\left[\hat{\boldsymbol{\Sigma}}\,\hat{\mathbf{R}}^2\right] = \mathrm{d}_\mathsf{x}^{-1}\,\mathrm{Tr}\left[(\hat{\mathbf{R}} - \lambda_t\hat{\mathbf{R}}^2)\right] \to_\mathbb{P} m_{t,1} - \lambda_t m_{t,2}.$$

Putting these together yields

$$\|\hat{\boldsymbol{\beta}}_t - \boldsymbol{\beta}_\star\|_2^2 \to_\mathbb{P} \lambda_t^2 m_{t,2} + \sigma_\varepsilon^2\gamma_t\left(m_{t,1} - \lambda_t m_{t,2}\right).$$

$$= \lambda_t m_{t,2}\left(\lambda_t - \sigma_\varepsilon^2\gamma_t\right) + \sigma_\varepsilon^2\gamma_t m_{t,1}.$$

Plugging this into the expression of $\mathcal{L}_t$ gives

$$\mathcal{L}_t \to_\mathbb{P} \sigma_\varepsilon^2 + \lambda_t m_{t,2}\left(\lambda_t - \sigma_\varepsilon^2\gamma_t\right) + \sigma_\varepsilon^2\gamma_t m_{t,1},$$

which concludes the proof for Proposition 3.

## C  Proof of Theorem 4

Recalling that $\tilde{\mathbf{y}} = \tilde{\mathbf{X}}\hat{\boldsymbol{\beta}}_t$ and $\hat{\boldsymbol{\beta}}_t = (\mathbf{X}^\top\mathbf{X} + \lambda_t n_t\mathbf{I}_{\mathrm{dx}})^{-1}\mathbf{X}^\top\mathbf{y}$, the vector $\hat{\boldsymbol{\beta}}_s$ can be written as

$$\hat{\boldsymbol{\beta}}_s = (\tilde{\mathbf{X}}^\top\tilde{\mathbf{X}} + \lambda_s n_s\mathbf{I}_{\mathrm{dx}})^{-1}\tilde{\mathbf{X}}^\top\tilde{\mathbf{y}}$$

$$= (\tilde{\mathbf{X}}^\top\tilde{\mathbf{X}} + \lambda_s n_s\mathbf{I}_{\mathrm{dx}})^{-1}\tilde{\mathbf{X}}^\top\tilde{\mathbf{X}}\,(\mathbf{X}^\top\mathbf{X} + \lambda_t n_t\mathbf{I}_{\mathrm{dx}})^{-1}\mathbf{X}^\top\mathbf{y}$$

$$= (\tilde{\mathbf{X}}^\top\tilde{\mathbf{X}} + \lambda_s n_s\mathbf{I}_{\mathrm{dx}})^{-1}\tilde{\mathbf{X}}^\top\tilde{\mathbf{X}}\,(\mathbf{X}^\top\mathbf{X} + \lambda_t n_t\mathbf{I}_{\mathrm{dx}})^{-1}\mathbf{X}^\top(\mathbf{X}\boldsymbol{\beta}_\star + \boldsymbol{\varepsilon}),$$

where we have used $\mathbf{y} = \mathbf{X}\boldsymbol{\beta}_\star + \boldsymbol{\varepsilon}$. Using the definition of the matrices $\tilde{\boldsymbol{\Sigma}}, \hat{\mathbf{R}}, \hat{\boldsymbol{\Sigma}}, \tilde{\mathbf{R}}$ from Section A, we can simplify the above expression as

$$\hat{\boldsymbol{\beta}}_s - \boldsymbol{\beta}_\star = (\tilde{\mathbf{R}}\tilde{\boldsymbol{\Sigma}}\hat{\mathbf{R}}\hat{\boldsymbol{\Sigma}} - \mathbf{I}_{\mathrm{dx}})\,\boldsymbol{\beta}_\star + n_t^{-1}\tilde{\mathbf{R}}\tilde{\boldsymbol{\Sigma}}\hat{\mathbf{R}}\mathbf{X}^\top\boldsymbol{\varepsilon}.$$

Hence, we have

$$\|\hat{\boldsymbol{\beta}}_s - \boldsymbol{\beta}_\star\|_2^2 = \boldsymbol{\beta}_\star^\top\left(\tilde{\mathbf{R}}\tilde{\boldsymbol{\Sigma}}\hat{\mathbf{R}}\hat{\boldsymbol{\Sigma}} - \mathbf{I}_{\mathrm{dx}}\right)^\top(\tilde{\mathbf{R}}\tilde{\boldsymbol{\Sigma}}\hat{\mathbf{R}}\hat{\boldsymbol{\Sigma}} - \mathbf{I}_{\mathrm{dx}})\,\boldsymbol{\beta}_\star + n_t^{-2}\boldsymbol{\varepsilon}^\top\mathbf{X}\hat{\mathbf{R}}\tilde{\boldsymbol{\Sigma}}\tilde{\mathbf{R}}^2\tilde{\boldsymbol{\Sigma}}\hat{\mathbf{R}}\mathbf{X}^\top\boldsymbol{\varepsilon} + o_\mathbb{P}(1)$$

$$= \mathrm{d}_\mathsf{x}^{-1}\,\mathrm{Tr}\left[(\tilde{\mathbf{R}}\tilde{\boldsymbol{\Sigma}}\hat{\mathbf{R}}\hat{\boldsymbol{\Sigma}} - \mathbf{I}_{\mathrm{dx}})^\top(\tilde{\mathbf{R}}\tilde{\boldsymbol{\Sigma}}\hat{\mathbf{R}}\hat{\boldsymbol{\Sigma}} - \mathbf{I}_{\mathrm{dx}})\right]$$

$$+ \sigma_\varepsilon^2\gamma_t\mathrm{d}_\mathsf{x}^{-1}\,\mathrm{Tr}\left[(\hat{\mathbf{R}}\hat{\boldsymbol{\Sigma}}\hat{\mathbf{R}})(\tilde{\boldsymbol{\Sigma}}\tilde{\mathbf{R}}^2\tilde{\boldsymbol{\Sigma}})\right] + o_\mathbb{P}(1),$$

where we have used the Hanson-Wright inequality and the facts that $\boldsymbol{\varepsilon}$ and $\boldsymbol{\beta}_\star$ are independent of other sources of randomness in the problem. Now, it remain to analyze the following traces in the high-dimensional proportional limit:

$$\tau_1 = \mathrm{d}_\mathsf{x}^{-1}\,\mathrm{Tr}\left[(\tilde{\mathbf{R}}\tilde{\boldsymbol{\Sigma}}\hat{\mathbf{R}}\hat{\boldsymbol{\Sigma}} - \mathbf{I}_{\mathrm{dx}})^\top(\tilde{\mathbf{R}}\tilde{\boldsymbol{\Sigma}}\hat{\mathbf{R}}\hat{\boldsymbol{\Sigma}} - \mathbf{I}_{\mathrm{dx}})\right], \quad \text{and}$$

$$\tau_2 = \mathrm{d}_\mathsf{x}^{-1}\,\mathrm{Tr}\left[(\hat{\mathbf{R}}\hat{\boldsymbol{\Sigma}}\hat{\mathbf{R}})(\tilde{\boldsymbol{\Sigma}}\tilde{\mathbf{R}}^2\tilde{\boldsymbol{\Sigma}})\right].$$

**Analysis of $\tau_1$.** Note that $\tilde{\mathbf{R}}\tilde{\boldsymbol{\Sigma}} = \mathbf{I}_{\mathrm{d_x}} - \lambda_s\tilde{\mathbf{R}}$ and $\hat{\mathbf{R}}\hat{\boldsymbol{\Sigma}} = \mathbf{I}_{\mathrm{d_x}} - \lambda_t\hat{\mathbf{R}}$, which gives

$$\tilde{\mathbf{R}}\tilde{\boldsymbol{\Sigma}}\hat{\mathbf{R}}\hat{\boldsymbol{\Sigma}} - \mathbf{I}_{\mathrm{d_x}} = -\lambda_s\tilde{\mathbf{R}} - \lambda_t\hat{\mathbf{R}} + \lambda_s\lambda_t\tilde{\mathbf{R}}\hat{\mathbf{R}}.$$

Plugging this into the expression for $\tau_1$, we arrive at

$$\tau_1 = \lambda_s^2\mathrm{d_x^{-1}Tr}\left[\tilde{\mathbf{R}}^2\right] + \lambda_t^2\mathrm{d_x^{-1}Tr}\left[\hat{\mathbf{R}}^2\right] + \lambda_s^2\lambda_t^2\mathrm{d_x^{-1}Tr}\left[\hat{\mathbf{R}}^2\tilde{\mathbf{R}}^2\right]$$

$$+ 2\lambda_s\lambda_t\mathrm{d_x^{-1}Tr}\left[\tilde{\mathbf{R}}\hat{\mathbf{R}}\right] - 2\lambda_s\lambda_t^2\mathrm{d_x^{-1}Tr}\left[\tilde{\mathbf{R}}\hat{\mathbf{R}}^2\right] - 2\lambda_s^2\lambda_t\mathrm{d_x^{-1}Tr}\left[\hat{\mathbf{R}}\tilde{\mathbf{R}}^2\right]$$

The limiting values of these traces can be computed as follows:

- **Term 1 and 2**: Let $\tilde{s}_k$ be the $k$-th eigenvalue of $\tilde{\boldsymbol{\Sigma}}$. We can use the arguments in Section A to write

$$\mathrm{d_x^{-1}Tr}\left[\tilde{\mathbf{R}}^2\right] \to_{\mathbb{P}} m_{s,2},$$

where $m_{s,2}$ is defined in Definition 2. Similarly, for the second term, we have

$$\mathrm{d_x^{-1}Tr}\left[\hat{\mathbf{R}}^2\right] \to_{\mathbb{P}} m_{t,2},$$

where the term $m_{t,2}$ is defined in Definition 2.

- **Term 3, 4, 5, and 6:** To analyze $\mathrm{d_x^{-1}Tr}\left[\hat{\mathbf{R}}^2\tilde{\mathbf{R}}^2\right]$, we can use the asymptotic freeness of independent Wishart random matrices [Voiculescu, 1991] (see also Capitaine and Donati-Martin [2007]), and the Stone-Weierstrass theorem to approximate the function $f(x) = (x + s)^{-2}$ using polynomials, to write

$$\mathrm{d_x^{-1}Tr}\left[\hat{\mathbf{R}}^2\tilde{\mathbf{R}}^2\right] = \left(\mathrm{d_x^{-1}Tr}\left[\hat{\mathbf{R}}^2\right]\right)\left(\mathrm{d_x^{-1}Tr}\left[\tilde{\mathbf{R}}^2\right]\right) + o_{\mathbb{P}}(1)$$

$$\to_{\mathbb{P}} m_{s,2}m_{t,2}.$$

Similarly, for the remaining terms, we have

$$\mathrm{d_x^{-1}Tr}\left[\hat{\mathbf{R}}\tilde{\mathbf{R}}\right] \to_{\mathbb{P}} m_{t,1}m_{s,1}$$

$$\mathrm{d_x^{-1}Tr}\left[\hat{\mathbf{R}}\tilde{\mathbf{R}}^2\right] \to_{\mathbb{P}} m_{t,1}m_{s,2}, \quad \text{and}$$

$$\mathrm{d_x^{-1}Tr}\left[\hat{\mathbf{R}}^2\tilde{\mathbf{R}}\right] \to_{\mathbb{P}} m_{t,2}m_{s,1}.$$

Putting these together, we arrive at the following conclusion:

$$\tau_1 \to_{\mathbb{P}} \lambda_t^2\, m_{t,2} + \lambda_s^2\, m_{s,2} + \lambda_s^2\lambda_t^2 m_{s,2}m_{t,2}$$

$$+ 2\lambda_t\lambda_s m_{t,1}m_{s,1} - 2\lambda_s\lambda_t^2 m_{s,1}m_{t,2} - 2\lambda_s^2\lambda_t m_{s,2}m_{t,1}.$$

**Analysis of $\tau_2$.** Again, we can use the identities $\tilde{\mathbf{R}}\tilde{\boldsymbol{\Sigma}} = \mathbf{I}_{\mathrm{d_x}} - \lambda_s\tilde{\mathbf{R}}$ and $\hat{\mathbf{R}}\hat{\boldsymbol{\Sigma}} = \mathbf{I}_{\mathrm{d_x}} - \lambda_t\hat{\mathbf{R}}$, and the asymptotic freeness of independent Wishart random matrices to write

$$\tau_2 = \mathrm{d_x^{-1}\,Tr}\left[(\hat{\mathbf{R}}\hat{\boldsymbol{\Sigma}}\hat{\mathbf{R}})(\tilde{\boldsymbol{\Sigma}}\tilde{\mathbf{R}}^2\tilde{\boldsymbol{\Sigma}})\right]$$

$$= \mathrm{d_x^{-1}Tr}\left[\left(\hat{\mathbf{R}} - \lambda_t\hat{\mathbf{R}}^2\right)\left(\mathbf{I}_{\mathrm{d_x}} - \lambda_s\tilde{\mathbf{R}}\right)^2\right]$$

$$= \mathrm{d_x^{-1}Tr}\left[\left(\hat{\mathbf{R}} - \lambda_t\hat{\mathbf{R}}^2\right)\right]\cdot\mathrm{d_x^{-1}Tr}\left[\left(\mathbf{I}_{\mathrm{d_x}} - \lambda_s\tilde{\mathbf{R}}\right)^2\right]$$

Hence, we can use an argument similar to the argument for the term 3 above to write

$$\tau_2 \to_{\mathbb{P}} \left(m_{t,1} - \lambda_t\, m_{t,2}\right)\left(1 + \lambda_s^2 m_{s,2} - 2\lambda_s m_{s,1}\right).$$

**Putting everything together.** Putting the conclusions above together, the limiting loss difference can be written as

$$\mathcal{L}_s - \mathcal{L}_t = \sigma_\varepsilon^2 + \tau_1 + \tau_2 + o_{\mathbb{P}}(1)$$

$$\to_{\mathbb{P}} (\sigma_\varepsilon^2 \gamma_t - \lambda_t) \left[ (m_{t,1} - \lambda_t m_{t,2}) \left( \lambda_s^2 m_{s,2} - 2\lambda_s m_{s,1} \right) \right] + \lambda_s^2 m_{s,2} \left( 1 - \lambda_t m_{t,1} \right) := \Delta$$

which completes the proof of the theorem.

## D  Proof of Theorem 5

From Theorem 4, we know that in the high-dimensional proportional limit, we have

$$\mathcal{L}_s - \mathcal{L}_t \to_{\mathbb{P}} \Delta = (\sigma_\varepsilon^2 \gamma_t - \lambda_t) \left[ (m_{t,1} - \lambda_t m_{t,2}) \left( \lambda_s^2 m_{s,2} - 2\lambda_s m_{s,1} \right) \right] + \lambda_s^2 m_{s,2} \left( 1 - \lambda_t m_{t,1} \right).$$

To study the $(\lambda_s, \lambda_t)$ pairs for which the strong model can outperform the teacher, we study the non-zero roots of the nonlinear equation $\Delta = 0$, which are the solutions to

$$(\sigma_\varepsilon^2 \gamma_t - \lambda_t) \left( \frac{m_{t,1} - \lambda_t m_{t,2}}{\lambda_t m_{t,1} - 1} \right) = \frac{\lambda_s m_{s,2}}{\lambda_s m_{s,2} - 2m_{s,1}} \tag{11}$$

where the left-hand side is a function of teacher parameters, and the right-hand side is a function of the student parameters.

**Right-Hand Side.** First, recall from Definition 2 that

$$m_{s,1} = m(\lambda_s; \gamma_s), \quad m_{s,2} = -\frac{\partial}{\partial \lambda} m(\lambda; \gamma)|_{\lambda_s, \gamma_s}.$$

in which

$$m(\lambda; \gamma) = -\frac{1}{2\gamma\lambda} \left[ 1 - \gamma + \lambda - \sqrt{(1 + \gamma + \lambda)^2 - 4\gamma} \right].$$

Thus, after simple Algebraic manipulations, we can write the right-hand side of (11) as

$$H_1 := \frac{\lambda_s m_{s,2}}{\lambda_s m_{s,2} - 2m_{s,1}}$$

$$= \frac{\lambda_s}{1 + \gamma_s^2 + 2\gamma_s(\lambda_s - 1) + \lambda_s + \lambda_s^2 + (1 - \gamma_s - \lambda_s)\sqrt{-4\gamma_s + (1 + \gamma_s + \lambda_s)^2}} \leq 0.$$

We plot $H_1$ as a function of $\lambda_s$ for different values of $\gamma_s$ in Figure 4. It can be seen that this function $H_1$ undergoes a phase transition at $\gamma_s = 1$. When $\gamma_s < 1$, the equation $H_1(\lambda, \gamma) = c$ for any $c \leq 0$ has only one solution for $\lambda_s$ and the function is strictly decreasing. However, for $\gamma_s > 0$, the function $H_1$ is non-monotone and $H_1(\lambda, \gamma) = c$ can have either none, one, or two solutions for $\lambda_s$. We will algebraically prove this fact below.

After simple algebraic manipulations, we find that setting $H_1(\lambda_s, \gamma_s) = c$ for some $c \leq 0$, we can have two potential solutions for $\lambda_s$ given by

$$\lambda_s^\pm(c, \gamma_s) = -\frac{1 + 2c + 5c^2 + 4c\gamma_s + 4c^2\gamma_s \pm (1 + 3c)\sqrt{(c - 1)^2 + 8c(1 + c)\gamma_s}}{4c(1 + c)}. \tag{12}$$

By inspecting the solutions, we find that when $\gamma_s \leq 1$, only the solution $\lambda_s^-$ is valid. However, when $\gamma_s > 1$, both $\lambda_s^+$ and $\lambda_s^-$ become valid solutions as long as

$$c < \frac{1}{1 - 4\gamma_s + \sqrt{\gamma_s(\gamma_s - 1)}}, \tag{13}$$

which ensures that $(c - 1)^2 + 8c(1 + c)\gamma_s \geq 0$.

Figure 5 shows the values of $\lambda_s$ for which $H_2(\lambda_s, \gamma_s) = c$ as a function of $c$, for different values of $\gamma_s$. The case of $\gamma_s = 1$ is shown with a blue dashed line. When $\gamma_s > 1$, two solutions can exist for $\lambda_s$. In this case, the largest $c$ for which two solutions exits are given by (13). However, for $\gamma_s < 1$, there is always one solution.

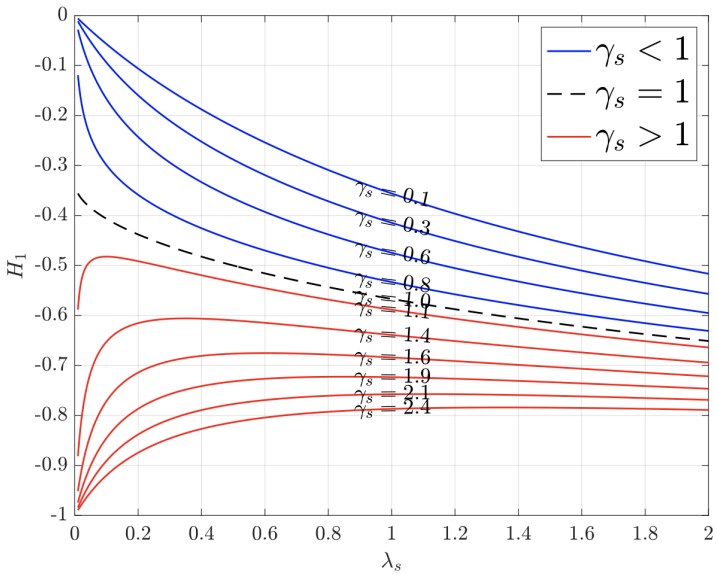

Figure 4: The function $H_1$, as a function of $\lambda_s$, for different values of $\gamma_s$. For the case with $\gamma_s > 1$, the equation $H_1(\lambda_s) = c$ with $c < 0$ can have two solutions. However, for $\gamma_s < 1$, there is always one solution.

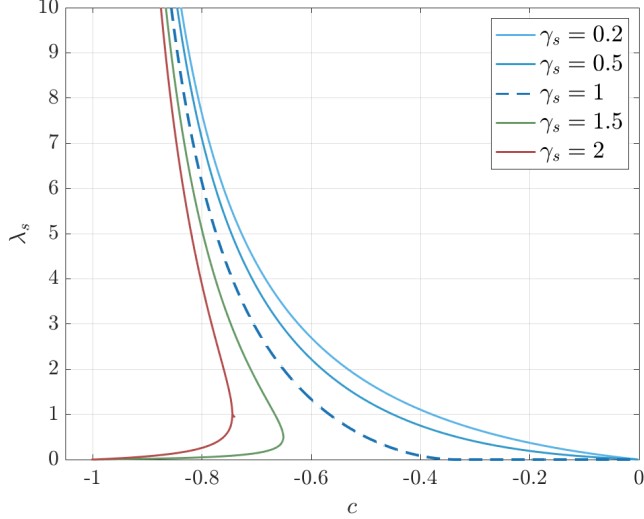

Figure 5: The parameters $\lambda_s$ for which $H_1(\lambda_s) = c$, for different values of $c$. Two solutions can exist when $\gamma_s > 1$. However, for $\gamma_s < 1$, only one solution can exist.

**Left-Hand Side.** We will now turn our attention to the left-hand side of (11). Recall that

$$m_{t,1} = m(\lambda_t; \gamma_t), \quad m_{t,2} = -\frac{\partial}{\partial \lambda} m(\lambda; \gamma)|_{\lambda_t, \gamma_t}.$$

We plug these into the right-hand side of (11), and after simplification, we arrive at

$$H_2 := (\sigma_\varepsilon^2 \gamma_t - \lambda_t) \left( \frac{m_{t,1} - \lambda_t m_{t,2}}{\lambda_t m_{t,1} - 1} \right) = \frac{\sigma_\varepsilon^2 \gamma_t - \lambda_t}{\sqrt{-4\gamma_t + (1 + \gamma_t + \lambda_t)^2}}.$$

Hence, $\sigma_\varepsilon^2 \gamma_t - \lambda_t$ determines the sign of $H_2$.

**Putting Everything Together.** After characterizing the functions $H_1$ and $H_2$, we can use these characterizations to prove the theorem.

- When $\sigma_\varepsilon^2 \gamma_t < \lambda_t$, we have $H_2 > 0$. Noting that $H_1 \leq 0$, we find that in this case, there is no solution for $\lambda_s$ such that $H_1 = H_2$.

- When $\sigma_\varepsilon^2 \gamma_t \geq \lambda_t$, we have $H_2 \leq 0$. Based on the analysis above for the right-hand side of (11), two cases can happen:

  - If $\gamma_s < 1$, we always have a solution $\bar{\lambda}$ given by $\lambda_s^-(c, \gamma_s)$ from (12) with

  $$c = \frac{\sigma_\varepsilon^2 \gamma_t - \lambda_t}{\sqrt{-4\gamma_t + (1 + \gamma_t + \lambda_t)^2}} \tag{14}$$

  such that $H_1 = H_2$.

  - If $\gamma_s > 1$, as long (13) holds; i.e.,

  $$c = \frac{\sigma_\varepsilon^2 \gamma_t - \lambda_t}{\sqrt{-4\gamma_t + (1 + \gamma_t + \lambda_t)^2}} \leq \frac{1}{1 - 4\gamma_s + \sqrt{\gamma_s(\gamma_s - 1)}},$$

  two solutions exists for $\lambda_s$ that satisfy for $H_1 = H_2$. The solutions are given by $(\lambda_s^-, \lambda_s^+)$ from (12). Consequently, we have $\Delta < 1$ as long as $\lambda_s \in (\lambda_s^-, \lambda_s^+)$.

These together finish the proof.

## E   Non-Monotone Error Curves

We use the theoretical prediction from Theorem 4 and plot the test error of the student model $\mathcal{L}_s$ as a function of $\gamma_t$. In Figure 6, we set the teacher regularizer to $\lambda_t \to 0$ (ridgeless regression) and set $\gamma_s = 0.1, \sigma_\varepsilon = 0.2$. We consider the same setting but with $\sigma_\varepsilon = 1$ in Figure 7. We observe that in both settings, the student has a non-monotone behavior for different values of $\lambda_s$, with a peak happening at the interpolation threshold $\gamma_t = 1$.

In Figure 8, we consider the same setting as Figure 7 with $\gamma_s = 0.1, \sigma_\varepsilon = 1$ and set $\lambda_t = \lambda_t^\star = \sigma_\varepsilon^2 \gamma_t$ (i.e., the optimal ridge regularizer). We observe that optimal regularization of the teacher model completely mitigates double descent in the student model; i.e., the test loss of the student model becomes monotone as a function of $\gamma_t$. This is in line with the findings of [Nakkiran et al., 2021] for standard ridge regression. Also, we observe that as predicted by Theorem 5, the student can never outperform the teacher.

In Figure 9, we consider the same setting as 8 but we set $\lambda_t = 0.15\lambda_t^\star$. We observe the in this setting where the teacher is still under-regularized, the student model still exhibits a non-monotone test error as a function of $\gamma_t$.

## F   Proof of Theorem 8

First, recall from Definition 8 that

$$\mathbf{\Gamma} = \mathbf{I}_{\mathrm{dx}} + \mathrm{d_x}\, \hat{\boldsymbol{\beta}}\hat{\boldsymbol{\beta}}^\top.$$

Using the Sherman-Morrison formula, we have

$$\mathbf{\Gamma}^{-1} = \mathbf{I}_{\mathrm{dx}} - \frac{\mathrm{d_x}\hat{\boldsymbol{\beta}}\hat{\boldsymbol{\beta}}^\top}{1 + \mathrm{d_x}\hat{\boldsymbol{\beta}}^\top\hat{\boldsymbol{\beta}}} = \mathbf{I}_{\mathrm{dx}} - (1 + o(1))\, \hat{\boldsymbol{\beta}}\hat{\boldsymbol{\beta}}^\top.$$

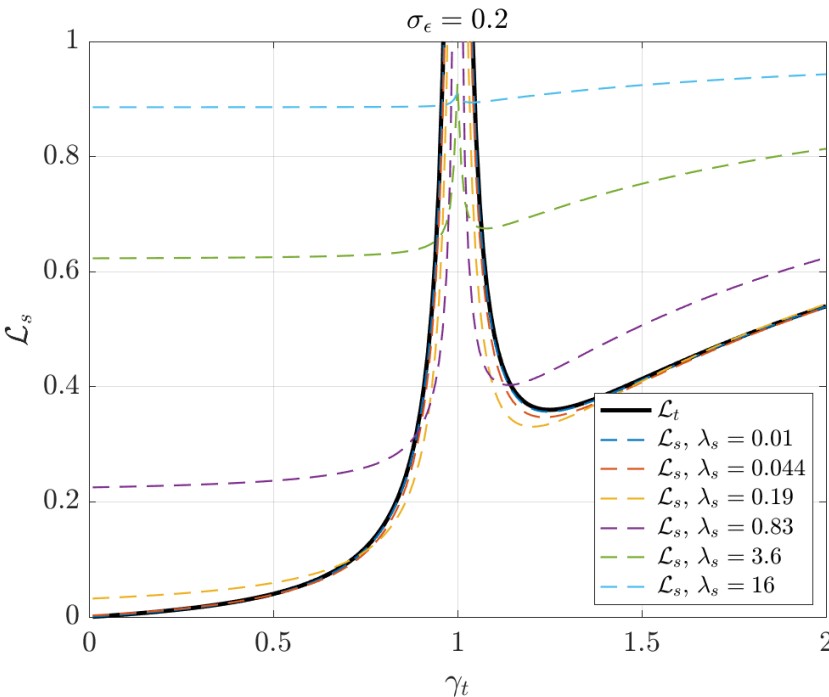

Figure 6: The test error of the student model $\mathcal{L}_s$ as a function of $\gamma_t$ for $\sigma_\varepsilon = 0.2, \gamma_s = 0.1, \lambda_t \to 0$ (ridgeless), and different values of $\lambda_s$.

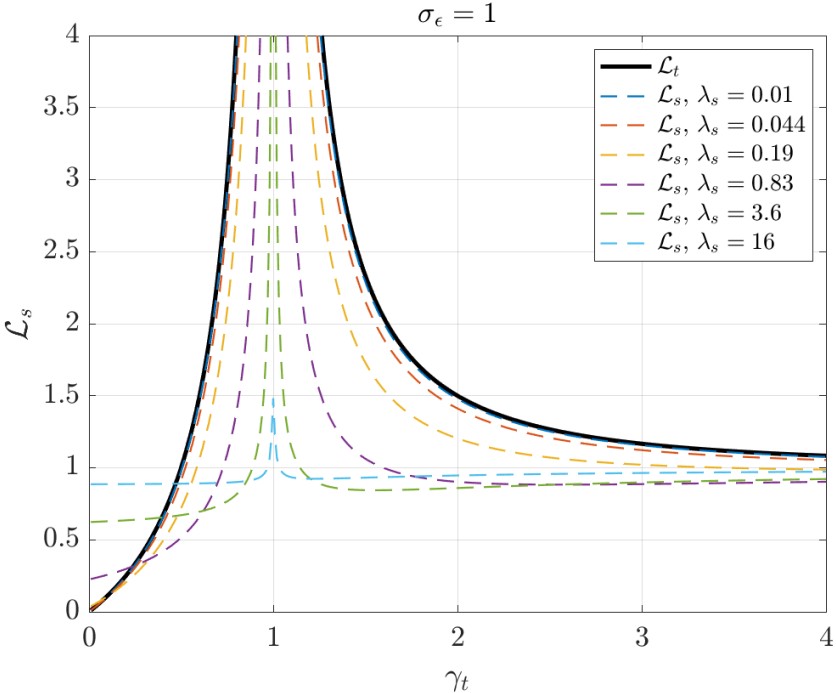

Figure 7: The test error of the student model $\mathcal{L}_s$ as a function of $\gamma_t$ for $\sigma_\varepsilon = 1, \gamma_s = 0.1, \lambda_t \to 0$ (ridgeless), and different values of $\lambda_s$.

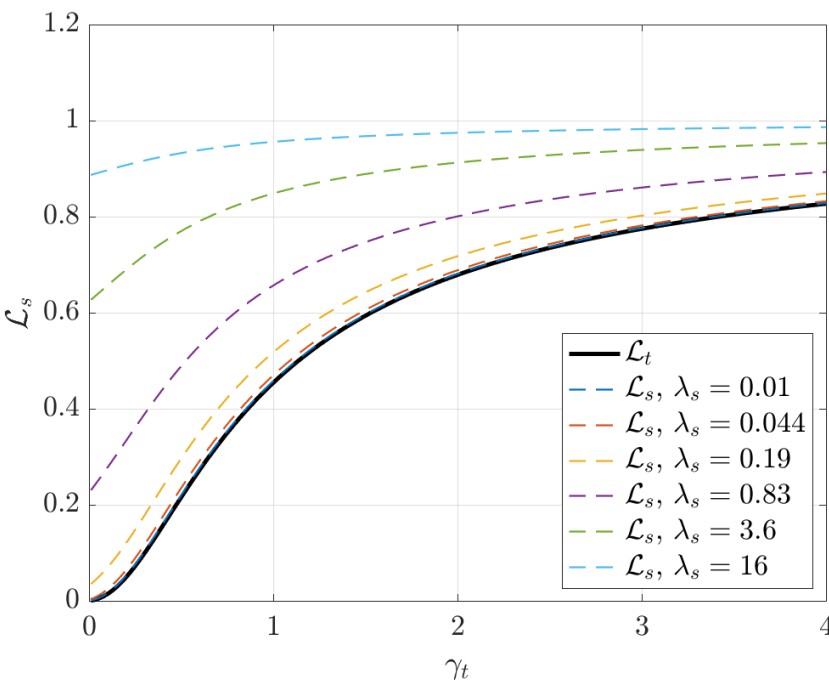

Figure 8: The test error of the student model $\mathcal{L}_s$ as a function of $\gamma_t$ for $\sigma_\varepsilon = 1, \gamma_s = 0.1, \lambda_t = \lambda_t^\star = \sigma_\varepsilon^2 \gamma_t$ (optimal ridge regularizer), and different values of $\lambda_s$.

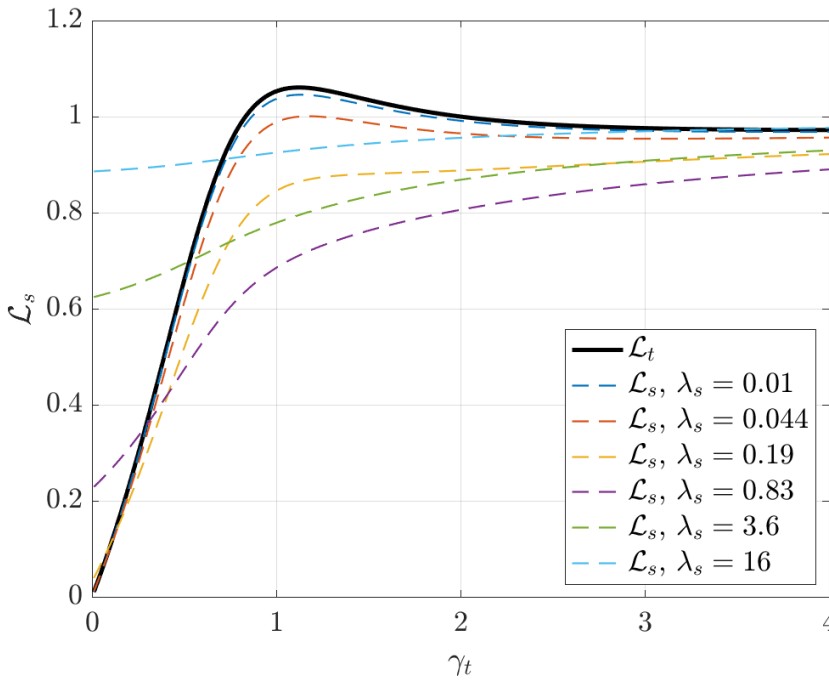

Figure 9: The test error of the student model $\mathcal{L}_s$ as a function of $\gamma_t$ for $\sigma_\varepsilon = 1, \gamma_s = 0.1, \lambda_t = 0.15\lambda_t^\star = 0.15\sigma_\varepsilon^2 \gamma_t$, and different values of $\lambda_s$.

In the setting of this theorem, we have $\hat{\boldsymbol{\beta}}_s = (\tilde{\boldsymbol{\Sigma}} + \lambda_s \boldsymbol{\Gamma}^{-1})^{-1} \tilde{\mathbf{X}}^\top \tilde{\mathbf{y}}/n_s$. Thus, we now focus on the generalized resolvent matrix $(\tilde{\boldsymbol{\Sigma}} + \lambda_s \boldsymbol{\Gamma}^{-1})^{-1}$. Using the Sherman-Morrison formula, this matrix can be expanded as

$$(\tilde{\boldsymbol{\Sigma}} + \lambda_s \boldsymbol{\Gamma}^{-1})^{-1} = (\tilde{\boldsymbol{\Sigma}} + \lambda_s \mathbf{I}_{\mathrm{d_X}} - \lambda_s \hat{\boldsymbol{\beta}}\hat{\boldsymbol{\beta}}^\top)^{-1} = \tilde{\mathbf{R}} + \frac{\lambda_s}{1 - \lambda_s \hat{\boldsymbol{\beta}}^\top \tilde{\mathbf{R}}\hat{\boldsymbol{\beta}}} \, \tilde{\mathbf{R}} \, \hat{\boldsymbol{\beta}}\hat{\boldsymbol{\beta}}^\top \tilde{\mathbf{R}}.$$

For simplicity, we define the scaler

$$\nu := \frac{\lambda_s}{1 - \lambda_s \hat{\boldsymbol{\beta}}^\top \tilde{\mathbf{R}} \hat{\boldsymbol{\beta}}}.$$

Hence, plugging the Sherman-Morrison expression back into the expression for $\hat{\boldsymbol{\beta}}_s$, we have

$$\hat{\boldsymbol{\beta}}_s = \left[ \tilde{\mathbf{R}} + \nu \, \tilde{\mathbf{R}} \, \hat{\boldsymbol{\beta}}\hat{\boldsymbol{\beta}}^\top \tilde{\mathbf{R}} \right] \tilde{\mathbf{X}} \tilde{\mathbf{y}}/n_s = \left[ \tilde{\mathbf{R}} + \nu \, \tilde{\mathbf{R}} \, \hat{\boldsymbol{\beta}}\hat{\boldsymbol{\beta}}^\top \tilde{\mathbf{R}} \right] \tilde{\boldsymbol{\Sigma}}\hat{\boldsymbol{\beta}}_t$$

$$= \left[ \tilde{\mathbf{R}} + \nu \, \tilde{\mathbf{R}} \, \hat{\boldsymbol{\beta}}\hat{\boldsymbol{\beta}}^\top \tilde{\mathbf{R}} \right] \tilde{\boldsymbol{\Sigma}} \left( \mathbf{R}\hat{\boldsymbol{\Sigma}}\boldsymbol{\beta}_\star + \mathbf{R}\mathbf{X}^\top \boldsymbol{\varepsilon}/n_t \right)$$

$$= \tilde{\mathbf{R}}\tilde{\boldsymbol{\Sigma}}\mathbf{R}\hat{\boldsymbol{\Sigma}}\boldsymbol{\beta}_\star + \tilde{\mathbf{R}}\tilde{\boldsymbol{\Sigma}}\mathbf{R}\frac{\mathbf{X}^\top \boldsymbol{\varepsilon}}{n_t} + \nu \left[ \tilde{\mathbf{R}}\hat{\boldsymbol{\beta}}\hat{\boldsymbol{\beta}}^\top \tilde{\mathbf{R}}\tilde{\boldsymbol{\Sigma}}\mathbf{R}\hat{\boldsymbol{\Sigma}}\boldsymbol{\beta}_\star + \tilde{\mathbf{R}}\hat{\boldsymbol{\beta}}\hat{\boldsymbol{\beta}}^\top \tilde{\mathbf{R}}\tilde{\boldsymbol{\Sigma}}\mathbf{R}\frac{\mathbf{X}^\top \boldsymbol{\varepsilon}}{n_t} \right].$$

We define $\mathbf{t}_1, \mathbf{t}_2, \mathbf{t}_3 \in \mathbb{R}^{\mathrm{d_X}}$ as

$$\mathbf{t}_1 = \tilde{\mathbf{R}}\tilde{\boldsymbol{\Sigma}}\mathbf{R}\hat{\boldsymbol{\Sigma}}\boldsymbol{\beta}_\star - \boldsymbol{\beta}_\star, \quad \mathbf{t}_2 = \tilde{\mathbf{R}}\tilde{\boldsymbol{\Sigma}}\mathbf{R}\frac{\mathbf{X}^\top \boldsymbol{\varepsilon}}{n_t},$$

$$\mathbf{t}_3 = \nu \left[ \tilde{\mathbf{R}}\hat{\boldsymbol{\beta}}\hat{\boldsymbol{\beta}}^\top \tilde{\mathbf{R}}\tilde{\boldsymbol{\Sigma}}\mathbf{R}\hat{\boldsymbol{\Sigma}}\boldsymbol{\beta}_\star + \tilde{\mathbf{R}}\hat{\boldsymbol{\beta}}\hat{\boldsymbol{\beta}}^\top \tilde{\mathbf{R}}\tilde{\boldsymbol{\Sigma}}\mathbf{R}\frac{\mathbf{X}^\top \boldsymbol{\varepsilon}}{n_t} \right].$$

With this definition, $\mathcal{L}_s = \sigma_\varepsilon^2 + \|\boldsymbol{\beta}_s - \boldsymbol{\beta}_\star\|_2^2$ can be written as

$$\mathcal{L}_s = \sigma_\varepsilon^2 + \|\mathbf{t}_1 + \mathbf{t}_2 + \mathbf{t}_3\|_2^2$$

$$= \|\mathbf{t}_1\|_2^2 + \|\mathbf{t}_2\|_2^2 + \|\mathbf{t}_3\|_2^2 + 2\mathbf{t}_1^\top \mathbf{t}_2 + 2\mathbf{t}_2^\top \mathbf{t}_3 + 2\mathbf{t}_1^\top \mathbf{t}_3.$$

We will analyze each term separately:

- The first and second terms $\|\mathbf{t}_1\|_2^2 + \|\mathbf{t}_2\|_2^2$ have already been calculated in in the proof of Theorem 4, we have

$$\|\mathbf{t}_1\|_2^2 + \|\mathbf{t}_2\|_2^2 - \mathcal{L}_t \to_{\mathbb{P}} \Delta,$$

  where $\Delta$ is defined in Theorem 4.

- For the third term, we can write

$$\|\mathbf{t}_3\|_2^2 = (\hat{\boldsymbol{\beta}}^\top \tilde{\mathbf{R}}^2 \hat{\boldsymbol{\beta}}) \cdot \left( \frac{\lambda_s}{1 - \lambda_s \hat{\boldsymbol{\beta}}^\top \tilde{\mathbf{R}} \hat{\boldsymbol{\beta}}} \right)^2 \cdot \left( \hat{\boldsymbol{\beta}}^\top \tilde{\mathbf{R}}\tilde{\boldsymbol{\Sigma}}\mathbf{R}\hat{\boldsymbol{\Sigma}}\boldsymbol{\beta}_\star \right)^2.$$

  From the definition of $m_{s,1}$ and $m_{s,2}$ and using the Marchenko-Pastur theorem, we have $\hat{\boldsymbol{\beta}}^\top \tilde{\mathbf{R}}^2 \hat{\boldsymbol{\beta}} \to_{\mathbb{P}} m_{s,2}$, and also $\hat{\boldsymbol{\beta}}^\top \tilde{\mathbf{R}} \hat{\boldsymbol{\beta}} \to_{\mathbb{P}} m_{s,1}$. Also, we can write

$$\hat{\boldsymbol{\beta}}^\top \tilde{\mathbf{R}}\tilde{\boldsymbol{\Sigma}}\hat{\mathbf{R}}\hat{\boldsymbol{\Sigma}}\boldsymbol{\beta}_\star = \zeta \mathrm{d_X}^{-1} \mathrm{Tr}\left( \tilde{\mathbf{R}}\tilde{\boldsymbol{\Sigma}}\hat{\mathbf{R}}\hat{\boldsymbol{\Sigma}} \right) = \zeta \, \mathrm{d_X}^{-1} \mathrm{Tr}\left( \tilde{\mathbf{R}}\tilde{\boldsymbol{\Sigma}} \right) \cdot \mathrm{d_X}^{-1} \mathrm{Tr}\left( \hat{\mathbf{R}}\hat{\boldsymbol{\Sigma}} \right)$$

$$\to_{\mathbb{P}} \zeta \cdot (1 - \lambda_s m_{s,1}) \cdot (1 - \lambda_t m_{t,1}),$$

  where $\zeta$ is defined in Assumption 7, and we have used the asymptotic freeness of independent Wishart random matrices [Voiculescu, 1991, Capitaine and Donati-Martin, 2007]. Hence,

$$\|\mathbf{t}_3\|_2^2 \to_{\mathbb{P}} \frac{\lambda_s^2 \zeta^2 m_{s,2}}{(1 - \lambda_s m_{s,1})^2}(1 - \lambda_s m_{s,1})^2(1 - \lambda_t m_{t,1})^2 = \lambda_s^2 \zeta^2 m_{s,2}(1 - \lambda_t m_{t,1})^2.$$

- For the fourth term, note that

$$2\mathbf{t}_1^\top \mathbf{t}_3 = 2((\tilde{\mathbf{R}}\tilde{\boldsymbol{\Sigma}}\hat{\mathbf{R}}\hat{\boldsymbol{\Sigma}} - \mathbf{I}_{\mathrm{d_X}})\hat{\boldsymbol{\Sigma}}\boldsymbol{\beta}_\star)^\top \tilde{\boldsymbol{\Sigma}}\hat{\mathbf{R}}\mathbf{X}^\top \boldsymbol{\varepsilon}/n_t \to_{\mathbb{P}} 0,$$

  using the Hanson-Wright inequality and the fact that $\boldsymbol{\varepsilon}$ is mean zero and independent of all other sources of randomness in the problem.

- Similarly to the fourth term, for the fifth term we can write

$$2\mathbf{t}_2^\top \mathbf{t}_3 = 2\nu \left( \tilde{\mathbf{R}} \tilde{\boldsymbol{\Sigma}} \hat{\mathbf{R}} \frac{\mathbf{X}^\top \boldsymbol{\varepsilon}}{n_t} \right)^\top \left[ \tilde{\mathbf{R}} \hat{\boldsymbol{\beta}} \hat{\boldsymbol{\beta}}^\top \tilde{\mathbf{R}} \tilde{\boldsymbol{\Sigma}} \hat{\mathbf{R}} \hat{\boldsymbol{\Sigma}} \boldsymbol{\beta}_\star + \tilde{\mathbf{R}} \hat{\boldsymbol{\beta}} \hat{\boldsymbol{\beta}}^\top \tilde{\mathbf{R}} \tilde{\boldsymbol{\Sigma}} \hat{\mathbf{R}} \frac{\mathbf{X}^\top \boldsymbol{\varepsilon}}{n_t} \right]$$

$$= 2\nu \left( \boldsymbol{\beta}^\top \tilde{\mathbf{R}} \tilde{\boldsymbol{\Sigma}} \hat{\mathbf{R}} \hat{\boldsymbol{\Sigma}} \boldsymbol{\beta}_\star \right) \left( n_t^{-1} \boldsymbol{\varepsilon}^\top \mathbf{X} \hat{\mathbf{R}} \tilde{\boldsymbol{\Sigma}} \tilde{\mathbf{R}}^2 \hat{\boldsymbol{\beta}} \right)$$

$$+ 2\nu \left( n_t^{-1} \hat{\boldsymbol{\beta}}^\top \tilde{\mathbf{R}} \tilde{\boldsymbol{\Sigma}} \hat{\mathbf{R}} \mathbf{X}^\top \boldsymbol{\varepsilon} \right) \left( n_t^{-1} \boldsymbol{\varepsilon}^\top \mathbf{X} \hat{\mathbf{R}} \tilde{\boldsymbol{\Sigma}} \tilde{\mathbf{R}}^2 \hat{\boldsymbol{\beta}} \right).$$

Using the Hanson-Wright inequality and the fact that $\boldsymbol{\varepsilon}$ is mean zero and independent of all other sources of randomness in the problem, we have $n_t^{-1} \boldsymbol{\varepsilon}^\top \mathbf{X} \hat{\mathbf{R}} \tilde{\boldsymbol{\Sigma}} \tilde{\mathbf{R}}^2 \hat{\boldsymbol{\beta}} \to_{\mathbb{P}} 0$. Hence,

$$2\mathbf{t}_2^\top \mathbf{t}_3 \to_{\mathbb{P}} 0.$$

- The sixth term can be expanded as follows:

$$2\mathbf{t}_1^\top \mathbf{t}_3 = 2\nu \left( \hat{\boldsymbol{\beta}}^\top \tilde{\mathbf{R}} \tilde{\boldsymbol{\Sigma}} \hat{\mathbf{R}} \hat{\boldsymbol{\Sigma}} \boldsymbol{\beta}_\star + n_t^{-1} \hat{\boldsymbol{\beta}}^\top \tilde{\mathbf{R}} \tilde{\boldsymbol{\Sigma}} \hat{\mathbf{R}} \mathbf{X}^\top \boldsymbol{\varepsilon} \right) \cdot \left( \hat{\boldsymbol{\beta}}^\top \tilde{\mathbf{R}} (\tilde{\mathbf{R}} \tilde{\boldsymbol{\Sigma}} \hat{\mathbf{R}} \hat{\boldsymbol{\Sigma}} - \mathbf{I}_{\mathrm{dx}}) \boldsymbol{\beta}_\star \right)$$

$$= 2\nu \left( \hat{\boldsymbol{\beta}}^\top \tilde{\mathbf{R}} \tilde{\boldsymbol{\Sigma}} \hat{\mathbf{R}} \hat{\boldsymbol{\Sigma}} \boldsymbol{\beta}_\star \right) \cdot \left( \hat{\boldsymbol{\beta}}^\top \tilde{\mathbf{R}}^2 \tilde{\boldsymbol{\Sigma}} \hat{\mathbf{R}} \hat{\boldsymbol{\Sigma}} \boldsymbol{\beta}_\star - \hat{\boldsymbol{\beta}}^\top \tilde{\mathbf{R}} \boldsymbol{\beta}_\star \right) + o_{\mathbb{P}}(1),$$

where again we have used the Hanson-Wright inequality and the fact that $\boldsymbol{\varepsilon}$ is mean zero and independent of all other sources of randomness in the problem. Above we have already shown above that

$$\hat{\boldsymbol{\beta}}^\top \tilde{\mathbf{R}} \tilde{\boldsymbol{\Sigma}} \hat{\mathbf{R}} \hat{\boldsymbol{\Sigma}} \boldsymbol{\beta}_\star \to_{\mathbb{P}} \zeta \cdot (1 - \lambda_s m_{s,1}) \cdot (1 - \lambda_t m_{t,1}).$$

With a similar argument, we have

$$\hat{\boldsymbol{\beta}}^\top \tilde{\mathbf{R}}^2 \tilde{\boldsymbol{\Sigma}} \hat{\mathbf{R}} \hat{\boldsymbol{\Sigma}} \boldsymbol{\beta}_\star = \zeta \cdot d_{\mathsf{x}}^{-1} \mathrm{Tr} \left( \tilde{\mathbf{R}}^2 \tilde{\boldsymbol{\Sigma}} \hat{\mathbf{R}} \hat{\boldsymbol{\Sigma}} \right) + o_{\mathbb{P}}(1)$$

$$= \zeta \cdot d_{\mathsf{x}}^{-1} \mathrm{Tr} \left( \tilde{\mathbf{R}}^2 \tilde{\boldsymbol{\Sigma}} \right) \cdot d_{\mathsf{x}}^{-1} \mathrm{Tr} \left( \hat{\mathbf{R}} \hat{\boldsymbol{\Sigma}} \right) + o_{\mathbb{P}}(1)$$

$$\to_{\mathbb{P}} \zeta (1 - \lambda_t m_{t,1}) \cdot (1 - \lambda_t m_{t,1}) \cdot (m_{s,1} - \lambda_s m_{s,2}).$$

Also, $\hat{\boldsymbol{\beta}}^\top \tilde{\mathbf{R}} \boldsymbol{\beta}_\star \to_{\mathbb{P}} \zeta m_{s,1}$. Hence, putting all together, we get

$$2\mathbf{t}_1^\top \mathbf{t}_3 \to_{\mathbb{P}} 2\zeta^2 \lambda_s (1 - \lambda_t m_{t,1}) \left[ \lambda_t \lambda_s m_{t,1} m_{s,2} - \lambda_t m_{t,1} m_{s,1} - \lambda_s m_{s,1} \right].$$

Thus, adding all the terms together, we have

$$\mathcal{L}_s - \mathcal{L}_t \to_{\mathbb{P}} \Delta - \zeta^2 \Delta_{\boldsymbol{\Gamma}}$$

where the expression for $\Delta$ is given in Theorem 4, and $\Delta_{\boldsymbol{\Gamma}}$ is given by

$$\Delta_{\boldsymbol{\Gamma}} := \lambda_s \left( -1 + \lambda_t m_{t,1} \right) \left[ -2\lambda_t m_{s,1} m_{t,1} + \lambda_s m_{s,2} (-1 + \lambda_t m_{t,1}) \right].$$

This concludes the proof.

## G  Proof of Proposition 10

The training loss for the teacher model is given by

$$\widehat{\mathcal{L}}_t := -\frac{1}{n_t} \sum_{i=1}^{n_t} y_i \hat{f}_t(\mathbf{x}_i) = -\frac{1}{n_t} \sum_{i=1}^{n_t} y_i \mathbf{a}_t^\top \sigma(\mathbf{W}_t \mathbf{x}_i).$$

Taking derivatives with respect to the matrix $\mathbf{W}_t$, we arrive at

$$\nabla_{\mathbf{W}_t} \widehat{\mathcal{L}}_t = -\frac{1}{n_t} \left[ (\mathbf{a}_t \mathbf{y}^\top) \odot \sigma'(\mathbf{W}_t \mathbf{X}^\top) \right] \mathbf{X}$$

Let $c_{\sigma,1}$ be the first Hermite coefficient of the activation function $\sigma$, and define $\sigma_\perp : \mathbb{R} \to \mathbb{R}$ as

$$\sigma_\perp(z) = \sigma(z) - c_{\sigma,1} z, \quad \forall z \in \mathbb{R},$$

where $\mathsf{E}_{z\sim\mathsf{N}(0,1)}[\sigma_\perp(z)] = 0$. Thus, we can write

$$\nabla_{\mathbf{W}_t}\widehat{\mathcal{L}}_t = -\frac{1}{n_t}\left[\left(\mathbf{a}_t\,\mathbf{y}^\top\right) \odot \left(c_{\sigma,1} + \sigma'_\perp(\mathbf{W}_{t,0}\mathbf{X}^\top)\right)\right]\mathbf{X}$$

$$= -\frac{c_{\sigma,1}}{n_t}\mathbf{a}_t\,\mathbf{y}^\top\mathbf{X} - \frac{1}{n_t}\left[\left(\mathbf{a}_t\,\mathbf{y}^\top\right) \odot \sigma'_\perp(\mathbf{W}_{t,0}\mathbf{X}^\top)\right]\mathbf{X}$$

By construction, the matrix $\sigma'_\perp(\mathbf{W}_{t,0}\mathbf{X}^\top)$ has mean zero entries. Thus, using [Vershynin, 2012, Theorem 5.44], we have $\|\sigma'_\perp(\mathbf{W}_{t,0}\mathbf{X}^\top)\|_{\mathrm{op}} = O(\sqrt{n_t})$. Hence, using Lemma 12, we have

$$\frac{1}{n_t}\left\|\left[\left(\mathbf{a}_t\,\mathbf{y}^\top\right)\odot\sigma'_\perp(\mathbf{W}_{t,0}\mathbf{X}^\top)\right]\mathbf{X}\right\|_{\mathrm{op}} = \frac{1}{n_t}\left\|\mathrm{diag}\left(\mathbf{a}_t\right)\sigma'_\perp(\mathbf{W}_{t,0}\mathbf{X}^\top)\,\mathrm{diag}(\mathbf{y})\mathbf{X}\right\|_{\mathrm{op}}$$

$$= \frac{1}{n_t}\cdot\frac{\mathrm{polylog}(p_t)}{\sqrt{p_t}}\cdot\sqrt{n_t}\cdot\mathrm{polylog}(n_t)\sqrt{n_t}$$

$$= \tilde{O}\left(\frac{1}{\sqrt{p_t}}\right),$$

where we have used the fact that $\|\mathbf{X}\|_{\mathrm{op}} = O(\sqrt{n_t})$ [Vershynin, 2012, Theorem 7.3.1], and the sub-gaussian maximal inequality to get $\|\mathbf{a}\|_\infty = p_t^{-1/2}\mathrm{polylog}(p_t)$. Similarly, using the sub-Weibull maximal inequality [Kuchibhotla and Chakrabortty, 2022, Proposition A.6 and Remark A.1], we have $\|\mathbf{y}\|_\infty = O(\mathrm{polylog}(n_t))$. As a result, for any $\boldsymbol{\beta}\in\mathbb{R}^{\mathrm{d_x}}$ with $\|\boldsymbol{\beta}\|_2 = 1$, we have

$$\|\nabla_{\mathbf{W}_t}\widehat{\mathcal{L}}_t\,\boldsymbol{\beta}\|_2 = n_t^{-1}\boldsymbol{\beta}^\top\mathbf{X}^\top\mathbf{y} + o_\mathbb{P}(1).$$

We will now study the case where $\boldsymbol{\beta}$ is the easy or the hard direction.

**Easy direction.** First, we let $\boldsymbol{\beta} = \boldsymbol{\beta}_e$. In this case, we have

$$\|\nabla_{\mathbf{W}_t}\widehat{\mathcal{L}}_t\,\boldsymbol{\beta}_e\|_2 = n_t^{-1}\boldsymbol{\beta}_e^\top\mathbf{X}^\top\left(\sigma_e(\mathbf{X}\boldsymbol{\beta}_e) + \sigma_h(\mathbf{X}\boldsymbol{\beta}_h)\right) + o_\mathbb{P}(1).$$

Note that $\mathbf{X}\boldsymbol{\beta}_e\in\mathbb{R}^{n_t}$ is a vector of i.i.d. $\mathsf{N}(0,1)$ entries. Thus, using the weak law of large numbers, we have

$$n_t^{-1}\boldsymbol{\beta}_e^\top\mathbf{X}^\top\sigma_e(\mathbf{X}\boldsymbol{\beta}_e) \to_\mathbb{P} \mathsf{E}_{z\sim\mathsf{N}(0,1)}[z\sigma_e(z)] = c_{\sigma_e,1}.$$

Also, recall the assumption that $\boldsymbol{\beta}_e$ and $\boldsymbol{\beta}_h$ are orthonormal vectors and $\mathbf{X}$ is a matrix with i.i.d. $\mathsf{N}(0,1)$ entries. Thus, $\mathbf{X}\boldsymbol{\beta}_e$ and $\mathbf{X}\boldsymbol{\beta}_h$ are independent and we have

$$n_t^{-1}\boldsymbol{\beta}_e^\top\mathbf{X}^\top\sigma_h(\mathbf{X}\boldsymbol{\beta}_h) \to_\mathbb{P} 0.$$

Thus, the gradient has a non-trivial alignment to the easy direction. Consequently, for $\widehat{\mathbf{W}}_t = \mathbf{W}_{t,1} - \eta_t\nabla_{\mathbf{W}_0}\widehat{\mathcal{L}}_t$, with $\eta_t = \Theta(1)$, we have $\|\widehat{\mathbf{W}}_t\boldsymbol{\beta}_e\|_{\mathrm{op}} \to_\mathbb{P} c > 0$, proving the first part of the proposition.

**Hard direction.** For the hard direction $\boldsymbol{\beta} = \boldsymbol{\beta}_h$, we have

$$\|\nabla_{\mathbf{W}_t}\widehat{\mathcal{L}}_t\,\boldsymbol{\beta}_h\|_2 = n_t^{-1}\boldsymbol{\beta}_h^\top\mathbf{X}^\top\left(\sigma_e(\mathbf{X}\boldsymbol{\beta}_e) + \sigma_h(\mathbf{X}\boldsymbol{\beta}_h)\right) + o_\mathbb{P}(1).$$

The first term $n_t^{-1}\boldsymbol{\beta}_h^\top\mathbf{X}^\top\sigma_e(\mathbf{X}\boldsymbol{\beta}_e)$ can be shown to be $o(1)$ with an argument identical to the argument above. For the second term, note that $\mathbf{X}\boldsymbol{\beta}_h\in\mathbb{R}^{n_t}$ is a vector with independent $\mathsf{N}(0,1)$ entries. Using the weak law of large numbers, we have

$$n_t^{-1}\boldsymbol{\beta}_h^\top\mathbf{X}^\top\sigma_h(\mathbf{X}\boldsymbol{\beta}_h) \to_\mathbb{P} \mathsf{E}_{z\mathsf{N}(0,1)}[z\sim\sigma_h(z)] = c_{\sigma_h,1} = 0,$$

where we have used the fact that the information exponent of $\sigma_h$ is lager than one; i.e., $c_{\sigma_h,1} = 0$. This shows that the gradient has no alignment to the hard direction, completing the proof.

## H   Proof of Theorem 11

From the proof of Proposition 10, we have

$$\widehat{\mathbf{W}}_t = \mathbf{W}_{t,0} + c_{\sigma,1}\eta_t\mathbf{a}_t\hat{\boldsymbol{\beta}}_e^\top + \boldsymbol{\Delta},$$

where $\hat{\boldsymbol{\beta}}_e = n_t^{-1}\mathbf{X}^\top \mathbf{y}$ and $\|\boldsymbol{\Delta}\|_{\mathrm{op}} = o(1)$. Given the fresh indepednent set of samples $\tilde{\mathbf{X}}$, the updated teacher model labels them as

$$\tilde{\mathbf{y}} = \tilde{\mathbf{F}}\mathbf{a}_t, \quad \text{with} \quad \tilde{\mathbf{F}} = \sigma(\tilde{\mathbf{X}}\widehat{\mathbf{W}}_t^\top) \in \mathbb{R}^{n_s \times p_t}$$

and the training loss for the student model given by

$$\widehat{\mathcal{L}}_s := -\frac{1}{n_s}\sum_{i=1}^{n_s}\tilde{y}_i\hat{f}_t(\tilde{\mathbf{x}}_i) = -\frac{1}{n_s}\sum_{i=1}^{n_s}\tilde{y}_i\mathbf{a}_s^\top\sigma(\mathbf{W}_s\tilde{\mathbf{x}}_i).$$

Taking derivatives with respect to the matrix $\mathbf{W}_t$, we arrive at

$$\nabla_{\mathbf{W}_s}\widehat{\mathcal{L}}_s\Big|_{\mathbf{W}_{s,0}} = -\frac{1}{n_s}\left[\left(\mathbf{a}_s\tilde{\mathbf{y}}^\top\right) \odot \sigma'(\mathbf{W}_{s,0}\tilde{\mathbf{X}}^\top)\right]\tilde{\mathbf{X}}. \tag{15}$$

To analyze the gradient, we should first characterize $\sigma'(\mathbf{W}_{s,0}\tilde{\mathbf{X}}^\top)$ and $\tilde{\mathbf{y}}$.

**Analysis of $\tilde{\mathbf{y}}$.** The feature matrix $\tilde{\mathbf{F}}$ is given by

$$\tilde{\mathbf{F}} = \sigma(\tilde{\mathbf{X}}\widehat{\mathbf{W}}_t^\top) = \sigma(\tilde{\mathbf{X}}\mathbf{W}_{t,0}^\top + c_{\sigma,1}\eta_t\tilde{\mathbf{X}}\hat{\boldsymbol{\beta}}_e\mathbf{a}_t^\top),$$

which is a nonlinear transform applied element-wise to a spiked random matrix. Following the recent results in nonlinear random matrix theory (e.g., Moniri et al. [2024], Wang et al. [2022], Moniri and Hassani [2024], Guionnet et al. [2023], Feldman [2025]), in the regime where $\eta_t = \Theta(1)$, we Hermite expand the nonlinearity as follows:

$$\tilde{\mathbf{F}} = \sigma\left(\tilde{\mathbf{X}}\widehat{\mathbf{W}}_t^\top\right) = \sigma\left(\tilde{\mathbf{X}}\mathbf{W}_{t,0}^\top + c_{\sigma,1}\eta_t\tilde{\mathbf{X}}\hat{\boldsymbol{\beta}}_e\mathbf{a}_t^\top\right)$$

$$= \sum_{k=1}^\infty c_{\sigma,k}H_k\left(\tilde{\mathbf{X}}\mathbf{W}_{t,0}^\top + c_{\sigma,1}\eta_t\tilde{\mathbf{X}}\hat{\boldsymbol{\beta}}_e\mathbf{a}_t^\top\right).$$

Using Lemma 15 element-wise, we can expand this matrix further

$$\tilde{\mathbf{F}} = \sum_{k=1}^\infty\sum_{j=0}^k\binom{k}{j}c_{\sigma,1}^j\eta_t^j c_{\sigma,k}H_{k-j}\left(\tilde{\mathbf{X}}\mathbf{W}_{t,0}^\top\right) \odot \left((\tilde{\mathbf{X}}\hat{\boldsymbol{\beta}}_e)^{\odot j}\mathbf{a}_t^{\odot j\top}\right)$$

$$= \sum_{k=1}^\infty c_{\sigma,k}H_k\left(\tilde{\mathbf{X}}\mathbf{W}_{t,0}^\top\right) + \sum_{k=1}^\infty c_{\sigma,1}^k\eta_t^k c_{\sigma,k}\left((\tilde{\mathbf{X}}\hat{\boldsymbol{\beta}}_e)^{\odot k}\mathbf{a}_t^{\odot k\top}\right)$$

$$+ \sum_{k=1}^\infty\sum_{j=1}^{k-1}\binom{k}{j}c_{\sigma,1}^j\eta_t^j c_{\sigma,k}H_{k-j}\left(\tilde{\mathbf{X}}\mathbf{W}_{t,0}^\top\right) \odot \left((\tilde{\mathbf{X}}\hat{\boldsymbol{\beta}}_e)^{\odot j}\mathbf{a}_t^{\odot j\top}\right).$$

Note that the first sum can be written as

$$\sum_{k=1}^\infty c_{\sigma,k}H_k\left(\tilde{\mathbf{X}}\mathbf{W}_{t,0}^\top\right) = \sigma(\tilde{\mathbf{X}}\mathbf{W}_{t,0}^\top).$$

In the second sum, by a simple sub-multiplicativity argument, the $k$-th term has an operator norm bounded by

$$\left\|c_{\sigma,1}^k\eta_t^k c_{\sigma,k}\left((\tilde{\mathbf{X}}\hat{\boldsymbol{\beta}}_e)^{\odot k}\mathbf{a}_t^{\odot k\top}\right)\right\|_{\mathrm{op}} = O\left(p_t^{1-k/2}\right)$$

which is $o(\sqrt{p_t})$ when $k > 1$. Moreover, using Lemma 12, the $(k, j)$-th term of the third sum has an operator upper bounded by

$$\left\|\binom{k}{j}c_{\sigma,1}^j\eta_t^j c_{\sigma,k}H_{k-j}\left(\tilde{\mathbf{X}}\mathbf{W}_{t,0}^\top\right) \odot \left((\tilde{\mathbf{X}}\hat{\boldsymbol{\beta}}_e)^{\odot j}\mathbf{a}_t^{\odot j\top}\right)\right\|_{\mathrm{op}}$$

$$= \left\|\binom{k}{j}c_{\sigma,1}^j\eta_t^j c_{\sigma,k}\,\mathrm{diag}\left((\tilde{\mathbf{X}}\hat{\boldsymbol{\beta}}_e)^{\odot j}\right)H_{k-j}\left(\tilde{\mathbf{X}}\mathbf{W}_{t,0}^\top\right)\mathrm{diag}\left(\mathbf{a}_t^{\odot j}\right)\right\|_{\mathrm{op}}$$

$$= \tilde{O}\left(p_t^{-j/2}\cdot n_s^{1/2}\right) = o(p_t^{1/2}).$$

Putting everything together, we have

$$\tilde{\mathbf{F}} = \sigma(\tilde{\mathbf{X}}\mathbf{W}_{t,0}^\top) + c_{\sigma,1}^2 \eta_t (\tilde{\mathbf{X}}\hat{\boldsymbol{\beta}}_e)\mathbf{a}_t^\top + \boldsymbol{\Delta},$$

where $\|\boldsymbol{\Delta}\|_{\mathrm{op}} = o\left(\sqrt{n_s}\right)$. Hence, recalling that $\|\mathbf{a}_t\|_2 = \Theta(1)$, we have

$$\tilde{\mathbf{y}} = \tilde{\mathbf{F}}\mathbf{a}_t = \sigma(\tilde{\mathbf{X}}\mathbf{W}_{t,0}^\top)\,\mathbf{a}_t + c_{\sigma,1}^2 \eta_t (\tilde{\mathbf{X}}\hat{\boldsymbol{\beta}}_e) + \boldsymbol{\delta} \tag{16}$$

in which $\|\boldsymbol{\delta}\|_2 = o(\sqrt{n_s})$.

**Derivative Term $\sigma'(\mathbf{X}\mathbf{W}_{s,0}^\top)$.**   Recall that we have $\mathbf{W}_{s,0} = \bar{\mathbf{W}}_{s,0} + \tau\,\bar{\mathbf{a}}\,\boldsymbol{\beta}_h^\top$. Hence,

$$\sigma'(\tilde{\mathbf{X}}\mathbf{W}_{s,0}^\top) = \sigma'(\tilde{\mathbf{X}}\bar{\mathbf{W}}_{s,0}^\top + \tau\,(\tilde{\mathbf{X}}\boldsymbol{\beta}_h)\,\bar{\mathbf{a}}^\top).$$

This is again a nonlinearity applied element-wise to a spiked random matrix. Similar to the argument for $\tilde{\mathbf{F}}$, we can use Lemma 15 to write

$$\sigma'(\tilde{\mathbf{X}}\mathbf{W}_{s,0}^\top) = \sum_{k=1}^\infty \sum_{j=0}^k \binom{k}{j}\tau^j c_{\sigma',k} H_{k-j}\left(\tilde{\mathbf{X}}\bar{\mathbf{W}}_{s,0}^\top\right) \odot \left((\tilde{\mathbf{X}}\boldsymbol{\beta}_h)^{\odot j}\,\bar{\mathbf{a}}^{\odot j\top}\right)$$

$$= \sum_{k=1}^\infty c_{\sigma',k} H_k\left(\tilde{\mathbf{X}}\bar{\mathbf{W}}_{s,0}^\top\right) + \sum_{k=1}^\infty \tau^k c_{\sigma',k}\left((\tilde{\mathbf{X}}\boldsymbol{\beta}_h)^{\odot k}\,\bar{\mathbf{a}}^{\odot k\top}\right)$$

$$+ \sum_{k=1}^\infty \sum_{j=1}^{k-1} \binom{k}{j}\tau^j c_{\sigma',k}\,H_{k-j}\left(\tilde{\mathbf{X}}\bar{\mathbf{W}}_{s,0}^\top\right) \odot \left((\tilde{\mathbf{X}}\boldsymbol{\beta}_h)^{\odot j}\bar{\mathbf{a}}^{\odot j\top}\right).$$

Similar to the reasoning used for $\tilde{\mathbf{F}}$, we have

$$\left\|\binom{k}{j}\tau^j c_{\sigma',k}\,H_{k-j}\left(\tilde{\mathbf{X}}\bar{\mathbf{W}}_{s,0}^\top\right) \odot \left((\tilde{\mathbf{X}}\hat{\boldsymbol{\beta}}_h)^{\odot j}\bar{\mathbf{a}}^{\odot j\top}\right)\right\|_{\mathrm{op}} = \tilde{O}\left(n_s^{1/2}\left(\frac{\tau}{\sqrt{p_s}}\right)^j\right)$$

which is $o(\sqrt{n_s})$ as long as $\tau = o(\sqrt{p_s})$. Thus, we have

$$\sigma'\left(\tilde{\mathbf{X}}\mathbf{W}_{s,0}^\top\right) = \sigma'\left(\tilde{\mathbf{X}}\bar{\mathbf{W}}_{s,0}^\top\right) + \sum_{k=1}^\infty \tau^k c_{\sigma',k}\left((\tilde{\mathbf{X}}\hat{\boldsymbol{\beta}}_h)^{\odot k}\,\bar{\mathbf{a}}^{\odot k\top}\right) + \bar{\boldsymbol{\Delta}} \tag{17}$$

in which $\|\bar{\boldsymbol{\Delta}}\|_{\mathrm{op}} = o(\sqrt{n_s})$.

**Gradient of the Loss.**   Now, we have all the ingredients to study the gradient of the loss function of the student model. Plugging (17) into (15), we have

$$\nabla_{\mathbf{W}_s}\widehat{\mathcal{L}}_s\Big|_{\mathbf{W}_{s,0}} = -\frac{1}{n_s}\left[(\mathbf{a}_s\,\tilde{\mathbf{y}}^\top) \odot \left[\sigma'\left(\bar{\mathbf{W}}_{s,0}\tilde{\mathbf{X}}^\top\right) + \sum_{k=1}^\infty \tau^k c_{\sigma',k}\left(\bar{\mathbf{a}}^{\odot k}\,(\tilde{\mathbf{X}}\hat{\boldsymbol{\beta}}_h)^{\odot k\top}\right) + \bar{\boldsymbol{\Delta}}\right]\right]\tilde{\mathbf{X}},$$

which we decompose as $\nabla_{\mathbf{W}_s}\widehat{\mathcal{L}}_s\Big|_{\mathbf{W}_{s,0}} = \mathbf{G}_1 + \mathbf{G}_2$ where $\mathbf{G}_1$ and $\mathbf{G}_2$ are defined as

$$\mathbf{G}_1 = -\frac{1}{n_s}\left[(\mathbf{a}_s\,\tilde{\mathbf{y}}^\top) \odot \sigma'\left(\bar{\mathbf{W}}_{s,0}\tilde{\mathbf{X}}^\top\right)\right]\tilde{\mathbf{X}},$$

$$\mathbf{G}_2 = -\frac{1}{n_s}\left[(\mathbf{a}_s\,\tilde{\mathbf{y}}^\top) \odot \left(\sum_{k=1}^\infty \tau^k c_{\sigma',k}\left(\bar{\mathbf{a}}^{\odot k}(\tilde{\mathbf{X}}\hat{\boldsymbol{\beta}}_h)^{\odot k\top}\right) + \bar{\boldsymbol{\Delta}}\right)\right]\tilde{\mathbf{X}}.$$

We will analyze each component separately.

**Analysis of $\mathbf{G}_1$.**   This term can be written as

$$\mathbf{G}_1 = -\frac{1}{n_s}\left[(\mathbf{a}_s\,\tilde{\mathbf{y}}^\top) \odot \sigma'\left(\bar{\mathbf{W}}_{s,0}\tilde{\mathbf{X}}^\top\right)\right]\tilde{\mathbf{X}}.$$

Let $c_{\sigma,1}$ be the first Hermite coefficient of the activation function $\sigma$, and define $\sigma_\perp : \mathbb{R} \to \mathbb{R}$ as

$$\sigma_\perp(z) = \sigma(z) - c_{\sigma,1}z, \quad \forall z \in \mathbb{R},$$

where $\mathsf{E}_{z\sim\mathsf{N}(0,1)}[\sigma_\perp(z)] = 0$. We can write write

$$\mathbf{G}_1 = -\mathbf{a}_s\left(\frac{\tilde{\mathbf{y}}^\top\tilde{\mathbf{X}}}{n_s}\right) - \frac{1}{n_s}\left[(\mathbf{a}_s\,\tilde{\mathbf{y}}^\top)\odot\sigma'_\perp\left(\bar{\mathbf{W}}_{s,0}\check{\mathbf{X}}^\top\right)\right]\tilde{\mathbf{X}}. \tag{18}$$

By using Lemma 12 and by a similar argument to the one used in the proof of Proposition 10, the operator norm of the second term of (18) be upper bounded as

$$\left\|\frac{1}{n_s}\left[(\mathbf{a}_s\,\tilde{\mathbf{y}}^\top)\odot\sigma'_\perp\left(\bar{\mathbf{W}}_{s,0}\check{\mathbf{X}}^\top\right)\right]\tilde{\mathbf{X}}\right\|_{\mathrm{op}} = o(1).$$

Using the characterization of $\tilde{\mathbf{y}}$ in (16), the first term of (18) can be written as $-\mathbf{a}_s\hat{\boldsymbol{\beta}}^\top$ with

$$\hat{\boldsymbol{\beta}} = \frac{1}{n_s}\tilde{\mathbf{X}}^\top\tilde{\mathbf{y}} = \frac{1}{n_s}\tilde{\mathbf{X}}^\top\left[\sigma(\tilde{\mathbf{X}}\mathbf{W}_{t,0}^\top)\mathbf{a}_t + c_{\sigma,1}^2\eta_t(\tilde{\mathbf{X}}\hat{\boldsymbol{\beta}}_e) + \boldsymbol{\delta}\right]$$

This vector aligns to the easy target direction:

$$\begin{aligned}
\boldsymbol{\beta}_e^\top\hat{\boldsymbol{\beta}} &= \frac{1}{n_s}(\tilde{\mathbf{X}}\boldsymbol{\beta}_e)^\top\left[\sigma(\tilde{\mathbf{X}}\mathbf{W}_{t,0}^\top)\mathbf{a}_t + c_{\sigma,1}^2\eta_t(\tilde{\mathbf{X}}\hat{\boldsymbol{\beta}}_e) + \boldsymbol{\delta}\right] \\
&= \frac{1}{n_s}(\tilde{\mathbf{X}}\boldsymbol{\beta}_e)^\top\left[\sigma(\tilde{\mathbf{X}}\mathbf{W}_{t,0}^\top)\mathbf{a}_t + c_{\sigma,1}^2\eta_t(\tilde{\mathbf{X}}\hat{\boldsymbol{\beta}}_e)\right] + o(1) \\
&= \frac{c_{\sigma,1}^2\eta_t}{n_s}(\tilde{\mathbf{X}}\boldsymbol{\beta}_e)^\top(\tilde{\mathbf{X}}\hat{\boldsymbol{\beta}}_e) + o(1) \to_\mathbb{P} c_e > 0. \tag{19}
\end{aligned}$$

**Analysis of $\mathbf{G}_2$.** To analyze this component, first note that we can write

$$\mathbf{G}_2 = -\left(\sum_{k=1}^\infty\frac{\tau^k c_{\sigma',k}}{n_s}\left((\mathbf{a}_s\odot\bar{\mathbf{a}}^{\odot k})\left(\tilde{\mathbf{y}}\odot(\tilde{\mathbf{X}}\hat{\boldsymbol{\beta}}_h)\right)^{\odot k\top}\right) + \bar{\boldsymbol{\Delta}}\right)\tilde{\mathbf{X}}.$$

The operator norm of the $k-$th term of the sum is given by

$$\left\|\left[\frac{\tau^k c_{\sigma',k}}{n_s}\left((\mathbf{a}_s\odot\bar{\mathbf{a}}^{\odot k})\left(\tilde{\mathbf{y}}\odot(\tilde{\mathbf{X}}\hat{\boldsymbol{\beta}}_h)\right)^{\odot k\top}\right) + \bar{\boldsymbol{\Delta}}\right]\tilde{\mathbf{X}}\right\|_{\mathrm{op}} = O\left(\left(\frac{\tau}{\sqrt{p_s}}\right)^k\right) = o(1),$$

where we have used the sub-multiplicativity of the operator norm, $\tau = o(\sqrt{p_s})$, and the fact that $\|\tilde{\mathbf{y}}\odot(\tilde{\mathbf{X}}\hat{\boldsymbol{\beta}}_h)\|_2 = \Theta(\sqrt{n_s})$ and $\|\mathbf{a}_s\odot\bar{\mathbf{a}}^{\odot k}\|_2 = O(p_s^{-k/2})$.

**Putting everything together.** Using the the results of the analyses above, we have

$$\widehat{\mathbf{W}}_s = \bar{\mathbf{W}}_{s,0} + \tau\,\bar{\mathbf{a}}\,\boldsymbol{\beta}_h^\top + \eta_s\mathbf{a}_s\hat{\boldsymbol{\beta}}^\top + \mathring{\boldsymbol{\Delta}}$$

where $\|\mathring{\boldsymbol{\Delta}}\|_{\mathrm{op}} = o(1)$. Hence, recalling (19), the updated weight matrix has non-vanishing correlation with both the easy and hard directions, completing the proof.

