# OpenReview forum: "On the Mechanisms of Weak-to-Strong Generalization: A Theoretical Perspective"
_NeurIPS.cc/2025/Conference — NeurIPS 2025 poster_

### Official Review · Reviewer_Rrob · 2025-06-01

**Clarity:** 3
**Significance:** 2
**Originality:** 3
**Rating:** 5
**Confidence:** 2

**Summary:**

This paper examines weak-to-strong generalization in several theoretical settings focused on ridge regression and multi-index regression. The authors identify a number of settings in which weak-to-strong generalization may occur, namely when the teacher is under-regularized, the student has partial access to ground truth weights via weighted ridge regularization, or the student has partial access to ground truth weights in a multi-index setup.

**Questions:**

Please see Strengths and Weaknesses above.

**Ethical Concerns:**

["NO or VERY MINOR ethics concerns only"]

**Final Justification:**

The author resolved my questions. Together with the other responses, this work seems to be a valuable contribution for this conference, and I recommend acceptance.

**Limitations:**

Yes

**Quality:**

3

**Strengths And Weaknesses:**

**Strengths**. The paper touches on a timely and fascinating topic. The results are precise. The writing is clear.

**Weaknesses**. I am likely not the target audience for this paper, so please take my comments with a hefty grain of salt. Please also defer to comments from reviewers who are more firmly placed in the target audience.

The results are impressively precise and the paper overall appears to be a nice application of increasingly popular RMT ideas, but I'm left unsure what to take away from this paper. The general finding seems to be that weak-to-strong generalization occurs when 1) the teacher is under-regularized for a ridge regression target (where regularization has a very concrete meaning), or 2) when some aspect of the target leaks to the student, either through a regularization weight or multi-index component.

For the first case, the result is quite interesting, though I'm unsure how to interpret the magnitude of difference. At best, for a very large regularization mismatch, the student outperforms the teacher by ~0.05. Is this a large or significant margin? Based on the plotted scale, this seems to be a fairly small difference. Should I interpret this result to mean the student can very slightly outperform the teacher when the teacher is virtually un-regularized? More broadly, what would regularization mean in an applied setting outside of ridge regression? Drawing from Burns et al. example you highlight as motivation, would GPT-2 be in some sense "under-regularized" for natural language, while GPT-4 compensates?

For the second case, it is perhaps unsurprising that a student would outperform the teacher if some aspect of the target function leaks to the student while remaining inaccessible to the teacher. Repeating the question from before, would you imagine that GPT-4 has access to some ground truth about natural language that remains out-of-reach for GPT-2, and for this reason succeeds at weak-to-strong generalization?

Overall, while the paper presents a nice theoretical argument, I'm unsure what the broader impact of these results are. Is there an actionable practice I can take away for performing weak-to-strong generalization on a real world task? Do we have a deeper understanding of weak-to-strong generalization beyond a linear setting or when the target leaks?

---

> ### Author Rebuttal · Authors · 2025-07-31
>
> We thank the reviewer for their review. We are glad that you found the paper timely, precise, and clear. Below, we will respond to each of the points raised in the review.
>
> - **Magnitude of difference.** In the paper, we focused on the conditions under which weak-to-strong generalization can happen. The exact magnitude of the improvement is dependent on the specific parameters of our model and can be smaller or larger, but the key takeaway is that a student can systematically improve upon a teacher by being regularized differently.
>
> - **Regularization in real scenarios.** We use ridge regression as a simple, analyzable proxy for the much more complex set of implicit and explicit regularizers in large neural networks. In a real-world scenario, the choice of architecture and optimization algorithm can also play a similar role. For example, if the student has an architecture that is better suited for the underlying tasks, e.g., it has some invariances that also exists in the target, it can correct some mistakes made by the weak teacher. This is very similar to setting 1 that was studied in the paper. Also recall that in Burns et al., it is seen that early stopping of the student is crucial for weak-to-strong generalization. Our paper propose a potential explanation for this phenomenon. A better inductive bias of the optimization algorithm can also play a similar role.
>
> - **GPT-4 has access to some ground truth about natural language that remains out-of-reach for GPT-2?** We hypothesize that a possible explanation of weak-to-strong generalization in GPT-4 and GPT-2 is that the tasks studied by Burns et al. [2024], e.g., the chess puzzles, have "leakage". The more extensive pre-training of GPT-4 (compared to GPT-2) on other tasks, combined with the fine-tuning on the data generated by GPT-2, can help in chess puzzles as well.
>
> - **Broader impact and take-aways of the paper.** Weak-to-strong generalization is a new generalization paradigm and is often considered a mysterious phenomenon and the mechanisms that enable it are not entirely clear. Our paper aims to demystify it. In this paper, we provide a theoretical grounding for weak-to-strong generalization. We show that weak-to-strong generalization is not unique to neural networks; even very simple linear models can exhibit it. We believe that this is an interesting finding that enables more in depth theoretical study of the phenomenon in a more controlled setting. We move beyond empirical observation to provide a rigorous theoretical analysis, identifying concrete and analyzable mechanisms—such as the student compensating for the teacher’s under-regularization or better leveraging pre-trained knowledge—that explain how and why a more capable student model can learn effectively from a weaker teacher.

---

> > ### Comment · Reviewer_Rrob · 2025-08-02
> >
> > Thanks for the detailed responses. This resolves my questions. Your work sounds like a valuable contribution, and I recommend acceptance. My score has been raised accordingly.

---

### Official Review · Reviewer_bbzK · 2025-06-16

**Clarity:** 3
**Significance:** 2
**Originality:** 2
**Rating:** 4
**Confidence:** 4

**Summary:**

This paper extends current research on weak-to-strong generalization by identifying three mechanisms that facilitate such phenomena: (1) compensation for teacher under-regularization in ridge regression; (2) better alignment of student regularization structures with the target function under weighted ridge regression; and (3) effective utilization of pretraining by the student in nonlinear settings.

**Questions:**

I have already raised concerns in the Weaknesses section，and none of them is very critical.

**Ethical Concerns:**

["NO or VERY MINOR ethics concerns only"]

**Final Justification:**

I acknowledge the contribution of this work and will keep my score to be positive.

**Limitations:**

yes

**Paper Formatting Concerns:**

No formatting concerns.

**Quality:**

3

**Strengths And Weaknesses:**

## Strengths:
1. The paper is well-written with clear logical flow and appropriate formalism.
2. Derivations are detailed and reflect strong theoretical proficiency.
3. The extension to nonlinear settings is interesting and adds depth to the study.


## Weaknesses:
1. In the nonlinear analysis, the definition of the *easy* link function appears to reduce the setting effectively to a linear case. This makes the learning of $\beta_e$ trivial.
2. If I understand correctly, in the nonlinear case (Theorem 11), the $\beta_h$ component in the student model originates entirely from pretraining, with early stopping controlling its degradation during subsequent training. However, even though the student learns $\beta_h$ while the teacher does not, if the constant $c$ in Proposition 10 is significantly larger than $c_e$ in Theorem 11, it is still unclear whether the student outperforms the teacher. If the presence of $\beta_h$ alone sufficed, the student would already outperform the teacher without any fine-tuning, which is not the intended conclusion.
3. Overall, while I consider the paper to meet the acceptance threshold, many of its key results and proofs hinge on the assumption that the student benefits from better pretraining while only modestly learning from the teacher’s supervision. As a result, the conclusions—though valid—are somewhat predictable, which limits the novelty of the contribution.

---

> ### Author Rebuttal · Authors · 2025-07-31
>
> We thank the reviewer for their review. We are glad that you found the analysis interesting, well-written, and detailed. Below, we will respond to each of the points raised in the review.
>
> - **Is learning the easy component trivial?** The easy component is a single-index function with unit information-exponent. It is known that Learning this function using SGD requires $\Theta(d)$ samples, compared to $\Omega(d)$ samples required for functions with an information-exponent greater than one. Thus, learning the easy component is still non-trivial (please see Ben-Arous et al. (2021) for more details).
>
> - **Weak-to-Strong generalization in setting 3.** We thank the reviewer for this comment. We note that the magnitude of the rank-one components in the updated weight matrices is controlled by the step-size of the gradient update. Thus, the step-size should be large enough for weak-to-strong generalization to happen. Also note that in the paper, we show that with $n = \Theta(d)$ samples, the updated weight matrix of the student will become aligned to the easy and hard components. With weights aligned with the target, even a very few $o(d)$ number of samples is enough for generalization. We will clarify these points in the final version of the paper.
>
> - **Novelty of the contributions.** We argue that we provide novel phenomenology that was not known in prior theoretical and empirical works on the subject. We show that weak-to-strong generalization is not unique to deep models and even very simple models can exhibit this phenomenon. We believe that this on its own is a solid contribution that enables the study of waek-to-strong generalization in a more controlled setting.  In setting 1 and 2, we study the case of ridge regression in detail. The analysis of the pairs $(\lambda_s, \lambda_t)$ for which weak-to-strong generalization happens is completely new and complements the prior work that do not study explicit regularization. In particular, we believe that the phase transition happens between under- and over-parameterized students, is very surprising and was not expected a priori. The analysis of setting 3 is also generalization of the results of Ba et al. [2022]. We show that the student model can learn the easy direction without "forgetting" the hard direction learned during pre-training. In fact, from a mathematical perspective, Theorem 11 is of independent interest and is actually a necessary component to study feature learning beyond the one step setup studied by Ba et al. [2022], Moniri et al. [2024], Cui et al. [2024], etc.

---

> > ### Comment · Reviewer_bbzK · 2025-08-01
> >
> > Thank you for your response, and I will keep my score to be positive.

---

### Official Review · Reviewer_cVVG · 2025-06-22

**Clarity:** 3
**Significance:** 2
**Originality:** 2
**Rating:** 4
**Confidence:** 3

**Summary:**

The paper addresses the question of when a student can surpass a teacher in isotropic-gaussian linear regression and a two-index model trained with a single step of gradient descent. In the linear case, the ability to surpass the teacher hinges on increasing regularization. In the two-index model, the student is initialized with a non-trivial alignment with the desired direction.

**Questions:**

- The statements of Proposition 10 and Theorem 11 appear strange when it comes to the constants. Are these absolute constants? Shouldn't $c_h$ depend on $\tau$?

- In figure 2, why did you increase the noise from the left and right figures? Is equation 5 not satisfied when $\sigma=1$?

**Ethical Concerns:**

["NO or VERY MINOR ethics concerns only"]

**Final Justification:**

Thank you for the response.

The main points that reflect the theory and experiments were addressed. There were also a few points in the rebuttal that I do not fully agree with, but these come down to semantics/storytelling and are not fundamental.

I have raised my score.

**Limitations:**

See the weaknesses section.

**Paper Formatting Concerns:**

The supplementary includes a version of the main paper that was updated after the original submission deadline (e.g., proposition 10 and the reference to section E after theorem 4 were updated). I gave the authors the benefit and checked the most up-to-date version (given in the supplementary).

**Quality:**

3

**Strengths And Weaknesses:**

Strengths:
- The paper is clearly written and easy to follow. The underlying topic is timely and well motivated.

- It is always nice when new observations are made in classical settings. I find Theorem 5, which clearly characterizes when the student can surpass the teacher in the linear setting (as long as equation 5 holds), to be a nice addition to the literature.

- The contour plots nicely complement the results.

Weaknesses:
- The isotropic-Gaussian setting is strict. By now, there are many papers analyzing linear regression in much more general settings.

- I'm not sure that the results of section 3 actually showcase weak to strong generalization, since the student is assumed to be initialized with weights that are already aligned with $\beta_h$. Training follows only a single gradient step, which prevents the student from changing significantly in the $\beta_h$ direction. This is fundamentally different from the case where the student achieves lower loss than the teacher despite being trained in a similar way.

- I believe the way the authors discuss the setting of section 3 in the abstract and introduction is misleading. Rather than saying things like "We assume that the easy component is highly specialized and task-specific, but, the hard component is a component shared across many tasks", it would be much more faithful to the theorems to state directly that the student weights are already aligned with the hard component. The storytelling regarding multiple tasks is not directly reflected in the math. It is fine to use this as motivation for the actual assumptions, but not to replace stating them.

---

> ### Author Rebuttal · Authors · 2025-07-31
>
> We thank the reviewer for their insightful comments. We are glad that you found our paper timely, well motivated, and a nice addition to the literature. Below, we respond to the points raised in the review:
>
> - **More general regression settings.** We agree with the reviewer that it is in principle possible to analyze this problem under more general assumptions on the input covariance. However, we note that the resulting expressions in this case will be very complex, and will not have a closed form; i.e., they will involve a fixed point equation. Thus, the results will become much harder to interpret. Moreover, we believe the study of the case of general covariance matrix is not central in our phenomenology of weak-to-strong generalization. In our view, studying the simplest possible settings that exhibit weak to strong generalization without additional clutter makes it easier to understand the fundamental mechanisms that enable it without unnecessary distractions.
>
> - **Results of section 3 and weak-to-strong generalization.** We thank the reviewer for this comment. We completely agree that setting 1, in which  the student and teacher are trained in a completely similar way, and the setting 3, where the weights of the student model are already aligned to some target components, are fundamentally different. But, we believe that setting 3 (similar to setting 1) actually does correspond to a form of weak-to-strong generalization. For instance, note that in Burns et al. [2024], the teacher model is GPT-2 and the student is GPT-4 and these model are indeed trained in a completely different way. GPT-4 already has learned many more capabilities in pretraining compared to GPT-2.  Also note that in even in the setting of Burns et al. [2024], the strong student should also be only fine-tuned (and not fully trained) on the outputs of the weak teacher, otherwise, the pre-trained capabilities of the strong student beyond that of the capabilities of the weak teacher will be eventually forgotten.
>
>
> - **Introduction.** We thank the reviewer for this comment. As suggested by the reviewer, we will adjust the introduction and explicitly mention our mathematical assumption alongside the motivation. Regarding the faithfulness of the motivation, we argue that the motivation of setting 3 is faithful to the mathematical assumption in the following way: the hard direction has been learned during the pre-training phase, and the network weights are already aligned to the hard direction, because it is a direction that is relevant for many different tasks and can be learned from them during pre-training. However, the easy direction is problem specific and could not be learned by the pre-training on other tasks. However, we will make sure that the actual mathematical assumption is also included  in the introduction.
>
> - **Constants.** The constants in Theorem 11 are not universal constants. The reviewer is right and $c_h$ does in indeed depend on the parameter $\tau$. We will adjust the notation in the final version to avoid this confusion.
>
> - **Choice of parameters in numerical experiments.** We solely made this decision for the choice of $\sigma$ to enhance visualization. We will also include the plot for $\sigma = 1$ in the final version of the paper.

---

> > ### Comment · Reviewer_cVVG · 2025-08-01
> > **Rebuttal (continued)**
> >
> > Thank you for the response.
> >
> > The main points that reflect the theory and experiments were addressed. There were also a few points in the rebuttal that I do not fully agree with, but these come down to semantics/storytelling and are not fundamental.
> >
> > I will raise my score.

---

### Official Review · Reviewer_V6rD · 2025-06-30

**Clarity:** 3
**Significance:** 3
**Originality:** 2
**Rating:** 5
**Confidence:** 3

**Summary:**

This paper studies the mechanisms of weak-to-strong generalization, on three different models, linear models trained using standard and weighted ridge regression, and generalized linear models with feature learning in the asymptotic regime. Their analysis includes both the over-parametrized and the under-parametrized regime and they provide explicit condition in which the student can outperform the teacher and when it fails to do so.

**Questions:**

**1.** Line 129, "that" should be changed to "than".

**2.** In lines 209-210, what does the statement "if $\gamma_s > 1$ then some information is lost" mean?

**3.** In line 267, doesn't taking $\zeta \rightarrow 0$ lead to recovering the setting in section 2.1? In that case, there are values of $\lambda_s$ for which the student can still outperform the teacher. I would appreciate the authors' comment on this.

**4.** In line 351, "teaching" should be changed to "teaches".

**Ethical Concerns:**

["NO or VERY MINOR ethics concerns only"]

**Final Justification:**

I find the paper well-written, organized, and its scope of analysis, comprehensive. The rebuttal has properly resolved the points I raised in the original review. I recommend acceptance.

**Limitations:**

See weaknesses and questions.

**Quality:**

3

**Strengths And Weaknesses:**

**Strengths**
Weak-to-strong generalization is a modern paradigm important to the improvement of performance in foundational models. This paper has provided precise conditions for linear models, when using a stronger regularization strength, the student's model's generalization error can outperform that of the teacher's. Furthermore, by analyzing the weighted ridge regression problem with a spiked covariance model, they show that the student can surpass the teacher if it has been adequately pre-trained. Finally on a generalized linear model which involves learning the two "easy" and "hard" directions, they showcase that applying one step of the gradient descent to the teacher's model can be still useful if the student has been sufficiently pre-trained.

Overall the paper is well-structured, well-written and easy to follow and their numerical experiments on synthetic supports their claims which align well with intuition.

**Weaknesses**

**1.** I believe that adding a discussion about the effect of the noise level improves the comprehensiveness of the paper. Namely, the question is as follows; given a budget on $\lambda_s$, what is the best the student can do in the face of various SNR levels. An study of low-SNR vs high-SNR regime is pertinent to the discussions presented in the main body.

**2.** The discussion around the exhibition of double-descent phenomenon in Appendix E should be included in the main body.

**3.** I recommend going beyond the synthetic and conducting experiments on real-world data such as MNIST and CIFAR-10 datasets.

I would like to add that I am open to increasing my scores in case the authors adequately address the points raised here.

---

> ### Author Rebuttal · Authors · 2025-07-31
>
> We thank the reviewer for their insightful comments. We are glad that you found our analysis of the weak-to-strong generalization in linear and nonlinear models interesting and well-structured. Below we respond to each of the comments raised in the review:
>
> - **Effect of noise level.**  We thank the reviewer for this constructive suggestion. To see the effect of the SNR, we first fix a teacher regularization parameter $\lambda_t$. As a corollary of Theorem 5, we know that in the high-SNR case where $\sigma_\epsilon^2 \leq \lambda_t/ \gamma_t$, weak to strong generalization cannot happen. However, the low-SNR case is completely different. In the low-SNR case ($\sigma_\epsilon^2 > \lambda_t/ \gamma_t$), if the student in under-parameterized, there is always a $\bar\lambda>0$ such that weak to strong generalization happens for $\lambda_s \in (0, \bar\lambda)$. Moreover, in the over-parameterized case $\gamma_s >1$, weak-to-strong generalization happens if we have
> $$
> \frac{\lambda_t}{\gamma_t} < \sigma_\epsilon^2 < \frac{\lambda_t}{\gamma_t} + \frac{1}{\gamma_t} \cdot \frac{\sqrt{(1 + \gamma_t + \lambda_t)^2 - 4 \gamma_t}}{4 \gamma_s + 4 \sqrt{\gamma_s^2 - \gamma_s}- 1.}
> $$
> This means that in the overparameterized case, weak-to-strong does not happen in the ultra-low SNR regime; i.e., in the regime with
> $$
>  \sigma_\epsilon^2 > \frac{\lambda_t}{\gamma_t} + \frac{1}{\gamma_t} \cdot \frac{\sqrt{(1 + \gamma_t + \lambda_t)^2 - 4 \gamma_t}}{4 \gamma_s + 4 \sqrt{\gamma_s^2 - \gamma_s}- 1}.
> $$
> We will discuss these in detail in the final version of the paper.
>
> - **Discussion of double descent.** In the final version, given the extra space, we will move the main parts of the discussion on double descent from the supplementary to the main body of the paper.
>
> - **Beyond synthetic datasets.** We thank the reviewer for this suggestion. From a theoretical side, we note that the limiting traces that appear in the expressions of our paper exhibit universality; i.e., the actual distribution of the of entries of the matrices does not matter and they can be replaced with a Gaussian matrix with matching first and second order moments and still the limit will be the same. This fact can be proved by a standard Lindeberg exchange argument. Thus, our results in theory do hold much more broadly than the Gaussian case. Experimentally, similar kinds of behavior is also seen with real datasets like MNIST. The new NeurIPS rebuttal rules for this year prevents us from sending new plots, but we will include a detailed discussion of this universality and numerical experiments in the final version of the paper.
>
>
> - **Information is lost.** In the case where $\gamma_s > 1$, we have $d > n_s$; i.e., the student model is over-parameterized. Information-theoretically, having access only to $n_s < d$ samples generated by the teacher, no algorithm can fully recover the $d$ dimensional vector $\hat\beta_t$. This is what we mean by "some information is lost". In the final version of the paper, we will clarify this point to avoid confusion in the final version of the paper.
>
>
> - **Limit of $\zeta \to 0$.** When $\zeta \to 0$, setting 2 reduces to  setting 1.  This can be seen by comparing the limiting expression of $\mathcal{L}_s - \mathcal{L}_t$ in Theorem 4 and 8. In setting 1, we have $\mathcal{L}_s - \mathcal{L}_t \to \Delta$ and is setting 2, we have
> $\mathcal{L}_s -   \mathcal{L}_t \to \Delta - \zeta^2 \Delta_g$
> where $\Delta_g \geq 0$. In particular, having $\zeta > 0$ enables the student to outperform the teacher in setups (i.e., values of $\gamma_s,\gamma_t, \dots$) in which a student model with $\zeta =0$ cannot.

---

> > ### Comment · Reviewer_V6rD · 2025-08-05
> >
> > I appreciate the authors' response to my questions. My concerns were addressed adequately and I have raised my score accordingly.

---

### Decision · Program_Chairs · 2025-09-17

**Decision:**

Accept (poster)

**Comment:**

The paper offers a simplified but compelling framework in which to study weak-to-strong generalization. Some of the insightful results that follow are clear conditions, such as alignment with the target, that compel a strong student to learn from a weak teacher, as well as a quantification of the behavior of a pre-trained strong student using its own strength (e.g., better features) to better learn from what information the teacher provides.

Reviewers generally appreciated the clarity of the presentation and significance of the results.There was some discussion on placing the novelty of the contributions more firmly within the literature. Some of the clarifications (about the effect of SNR and about limitations such as beyond regression or beyond synthetic data) should also be part of the paper. Otherwise, the paper is likely to be well-appreciated by the community.